# The biophysics, ecology, and biogeochemistry of functionally diverse, vertically- and horizontally-heterogeneous ecosystems: the Ecosystem Demography Model, version 2.2 — Part 2: Model evaluation for tropical South America

Marcos Longo[1,2,3], Ryan G. Knox[4,5], Naomi M. Levine[6], Abigail L. S. Swann[7], David M. Medvigy[8], Michael C. Dietze[9], Yeonjoo Kim[10], Ke Zhang[11], Damien Bonal[12], Benoit Burban[13], Plínio B. Camargo[14], Matthew N. Hayek[1,15], Scott R. Saleska[16], Rodrigo da Silva[17], Rafael L. Bras[18], Steven C. Wofsy[1], and Paul R. Moorcroft[1]

[1]Harvard University, Cambridge, MA, United States
[2]Embrapa Agricultural Informatics, Campinas, SP, Brazil
[3]Jet Propulsion Laboratory, California Institute of Technology, Pasadena, CA, United States
[4]Massachusetts Institute of Technology, Cambridge, MA, United States
[5]Lawrence Berkeley National Laboratory, Berkeley, CA, United States
[6]University of Southern California, Los Angeles, CA, United States
[7]University of Washington, Seattle, WA, United States
[8]University of Notre Dame, Notre Dame, IN, United States
[9]Boston University, Boston, MA, United States
[10]Department of Civil and Environmental Engineering, Yonsei University, Seoul 03722, Republic of Korea
[11]Hohai University, Nanjing, Jiangsu, China
[12]INRA, UMR 1137 EEF, 54280, Champenoux, France
[13]INRA, UMR 0745 EcoFoG, Campus Agronomique, 97379, Kourou, France
[14]University of São Paulo, Piracicaba, SP, Brazil
[15]New York University, New York, NY, United States
[16]University of Arizona, Tucson, AZ, United States
[17]Federal University of Western Pará, Santarém, PA, Brazil
[18]Georgia Institute of Technology, Atlanta, GA, United States

**Correspondence:** M. Longo
(mlongo@post.harvard.edu)

**Abstract.** The Ecosystem Demography Model version 2.2 (ED-2.2) is a terrestrial biosphere model that simulates the biophysical, ecological and biogeochemical dynamics of vertically and horizontally heterogeneous terrestrial ecosystems. In a companion paper (Longo et al., 2019), we described how the model solves the energy, water, and carbon cycles, and verified the high degree of conservation of these properties in long-term simulations that include long-term (multi-decadal) vegetation dynamics. Here, we present a detailed assessment of the model's ability to represent multiple processes associated with the biophysical and biogeochemical cycles, in Amazon forests. We use multiple measurements from eddy covariance towers, forest inventory plots and regional remote-sensing products to assess the model's ability to represent biophysical, physiological, and ecological processes at multiple time scales, ranging from sub-daily to century-long. The ED-2.2 model accurately describes the vertical distribution of light, water fluxes and the storage of water, energy and carbon in the canopy air space, the regional

distribution of biomass in tropical South America, and the variability of biomass as a function of environmental drivers. In addition, ED-2.2 qualitatively captures several emergent properties of the ecosystem found in observations, specifically observed relationships between above-ground biomass, mortality rates and wood density; however, the slopes of these relationships were not accurately captured. We also identified several limitations, including the model's tendency to overestimate the magnitude and seasonality of heterotrophic respiration, and to overestimate growth rates in a nutrient-poor tropical site. The evaluation presented here highlights the potential of incorporating structural and functional heterogeneity within biomes in ESMs, and to realistically represent their impacts on energy, water, and carbon cycles. We also identify several priorities for further model development.

## 1 Introduction

Terrestrial ecosystems are fundamental components of the earth system. Current estimates suggest that net terrestrial biosphere carbon uptake is near $3.2\,\mathrm{GtC\,yr^{-1}}$ (equivalent to $30\%$ of fossil fuel carbon emissions), albeit this sink is partially offset by land use change ($1.5\,\mathrm{GtC\,yr^{-1}}$), or $16\%$ of fossil fuel carbon emissions (Le Quéré et al., 2018). Climate change, land use change, and increase in atmospheric $CO_2$ are likely to alter the ability of terrestrial ecosystems to accumulate carbon, yet the magnitude and even the persistence of terrestrial biosphere sink, as predicted by dynamic global vegetation models (DGVMs) remains highly uncertain (Friedlingstein et al., 2006, 2014; Krause et al., 2018). The response of tropical forests to global change is of particular interest, because these ecosystems store nearly $40\%$ of the total terrestrial biomass (Erb et al., 2018), and their simulated response to climate change is strongly dominated by vegetation dynamics (Ahlström et al., 2015).

DGVMs have evolved considerably over the past three decades: most already contain detailed, mechanistic representations of biophysical and biogeochemical cycles (Fisher et al., 2018). However, two important factors limit the ability to reduce the uncertainty in DGVM predictions. First, observations of key biophysical and ecological variables (e.g. forest inventory plots and eddy covariance towers) are not evenly distributed, and tropical ecosystems are heavily under-sampled (Schimel et al., 2015). Second, DGVMs tend to describe ecosystems based on simple aggregate ecosystem characteristics that precludes proper representation of the heterogeneous environment in which individual plants live. In reality, the dynamics of an ecosystem is an emergent property that integrates the contributions of a system comprised of individuals with different strategies and abilities to access resources needed for their growth, survival, and reproduction. The emergent nature of the large-scale properties of terrestrial ecosystems has important consequences for how terrestrial ecosystems respond to climate change and other forms of environmental perturbation. (Moorcroft, 2003, 2006; Evans, 2012; Fisher et al., 2018).

In a companion paper (Longo et al., 2019), we described the Ecosystem Demography Model version 2.2 (ED-2.2). Unlike most DGVMs, the ED-2.2 model is a terrestrial biosphere model that characterizes and represents the vertical and functional structure of plant communities, as well as the horizontal heterogeneity of terrestrial ecosystems. Importantly, ED-2.2 has complete biophysical and biogeochemical cores that solve the energy, water, and carbon cycles of the vertically and horizontally heterogeneous plant canopy at sub-daily resolution. The biophysical and biogeochemical cores build on previous model devel-

opments by other DGVMs; however, in ED-2.2, we also account for the local variability of structure and composition in plant communities.

In this paper, we present a detailed evaluation of the ED-2.2 model, with focus on the multiple processes that comprise the biophysical and biogeochemical cores. Formal optimization of model parameters was beyond the scope of this paper, although it has been previously carried out with ED-2 (e.g. Medvigy et al., 2009; Kim et al., 2012; Fer et al., 2018). Instead, we conduct a detailed evaluation of the model in the Amazon region, using two data-rich sites with decadal-long series of measurements that quantify several water, energy, and carbon components fluxes and storage terms predicted by ED-2.2. We also evaluate the model's ability to represent both the regional distribution of biomass and forest structure across tropical South America, and the mechanisms that drive the spatial and temporal variability in carbon stocks and structure, by comparing the model results with independent field measurements and remote-sensing estimates. Together, these analyses establish the model's consistency and its ability to be applied in both short-term and long-term studies of terrestrial ecosystem dynamics.

## 2  Methods

Since its conception (Moorcroft et al., 2001; Hurtt et al., 2002), the plant community in the Ecosystem Demography model (ED) is represented by a hierarchical structure that accounts for both abiotic (e.g. soil texture and aspect) and biotic (e.g. age since last disturbance and disturbance types) effects on the vertical structure of canopy and the horizontal heterogeneity of such structures across the landscape. To characterize this sub-grid heterogeneity within the areas of interest and the changes of structure over time, ED solves a system of partial differential equations that describe growth, mortality and recruitment of individuals, as well as the changes in the plant community following disturbances. The current version of the model, ED-2.2, which is presented in a companion paper (Longo et al., 2019) builds on the developments by Medvigy et al. (2009) to quantify in detail the biophysical and biogeochemical cycles of energy, water, and carbon that fully accounts for the sub-grid heterogeneity of the above- and below-ground structure of the ecosystem.

In this manuscript, we evaluate a range of short-term (hours to seasons) biophysical and biogeochemical processes and long-term (years to centuries) demographic and ecological processes, with focus in tropical South America. Specifically, for the short-term processes, we used detailed information from eddy covariance tower sites in the Amazon to assess the model's ability to represent the canopy radiation transfer, evapotranspiration, sensible heat flux, temperature and water in sub-canopy pools (e.g. canopy air space and soils), and carbon fluxes (gross primary productivity and respiration). For the long-term processes, we evaluated the model ability to represent the distribution of biomass and leaf area index across tropical South America, and relevant demographic characteristics such as the simulated forest structure and demographic rates (e.g. mortality, growth). Importantly, we also tested the ecosystem response to environmental controls both for short- and long-term processes. Examples of such evaluations for short-term processes include the predicted response of gross primary productivity and respiration to environmental factors such as incoming radiation and soil moisture. For long-term processes, we tested how predicted carbon stocks were modulated by the regional variation of environmental controls such as rainfall and radiation, as well as emergent

ecosystem properties such as the average mortality rate and average wood density, both of which are also predicted by the
75 model.

## 2.1 Assessment of short-term fluxes

For most evaluations of biophysical and biogeochemical cycles, we ran ED-2.2 for two sites in the Amazon where both eddy
flux towers and forest inventories were available for a long period: the Guyaflux tower ($5°17'$N; $52°55'$W) at Paracou, French
Guiana (GYF; Bonal et al., 2008), and the Tapajós National Forest site ($2°51'$S; $54°58'$W), located in central Amazon (TNF;
Hutyra et al., 2008; Pyle et al., 2008). Both data sets underwent multiple-stage quality control. We used the meteorological
variables measured at the eddy covariance tower sites to drive these ED-2.2 simulations. These variables included tempera-
ture, specific humidity, pressure, wind speed, incoming solar radiation, incoming longwave radiation and precipitation. All
meteorological variables used as inputs for ED-2.2 were gap filled, following Longo (2014). Net ecosystem exchange (NEE)
was processed using the approach by Hayek et al. (2018b), which corrects flux bias due to lack of turbulence and turbulent-
85 independent divergence of $CO_2$.

To ensure that model and observations at or near eddy covariance flux towers could be directly compared, and that the
observed signal was strongly related to actual environment conditions, we saved model outputs every hour, and only used the
model output for the hours when each variable of interest was measured. Gross primary productivity (GPP) and ecosystem
respiration ($\dot{R}_{\mathrm{Eco}}$) are not measured but statistically modeled from NEE, therefore we compared all times in which NEE could
be estimated from tower observations. We also required that the 24-hour period preceding any given time had less than 24 gap
filled values among all seven driver variables. Finally, in ED-2.2, we solve fluxes for each local plant community (patch) within
the area of interest (polygon). Since eddy covariance towers provide flux measurements and estimates that are representative
of the entire plant community, to assess ED-2.2 results against the eddy covariance fluxes, we compared the tower estimates
with the ED-2.2 polygon averages (i.e. area-weighted average of fluxes for all simulated patches).
To evaluate the in-canopy radiation profile, we compared model results against measured profiles of photosynthetically active
radiation at two sites in the Brazilian Amazon: Jaru Biological Reserve (RJA: $10°05'$S; $61°56'$W) near Ji-Paraná, and Adolpho
Ducke Forest Reserve (MDK: $2°57'$S; $59°55'$W), near Manaus (Cabral et al., 1996; Tomasella et al., 2008). At RJA, data were
collected during two measurements campaigns as part of the Anglo-Brazilian Climate Observation Study (ABRACOS; Gash
et al., 1996) in August–September 1992 and April–June 1993 and measurements were taken at 2.3, 6.1, 11.6, 15.7, 21.3, and
100 35.0 m above ground; at MDK, the measurement campaign occurred in July–August 1991 and sensors were installed at 5.0,
10.0, 15.0, 20.0, 25.0, and 35.0 m above ground. In both cases two to three sensors were installed in each level either East or
West of the tower, and both sites used the same quantum radiation sensors of model SKP 215, skye Instruments Ltd, Powys,
UK (Tomasella et al., 2008). To reduce the impact of the towers on the measurements, we only used data from sensors east
of the towers in the morning and west of the towers in the afternoon (Mercado et al., 2007). The model was run for the same
105 days of year as the data. However, we simulated different years — 1999-2006 for MDK and 1999-2002 for RJA — using
the Large-Scale Biosphere-Atmosphere Experiment in Amazonia Data Model Intercomparison Project (LBA-MIP) data (de
Gonçalves et al., 2013), because we did not have data from all meteorological variables needed by ED-2.2 during the period

when the radiation profile data were collected. Also, the diurnal cycle of any point measurement within the canopy depends on local heterogeneities and can be dramatically affected by the Sun's azimuth and zenith angles (e.g. sun flecks when the sensor is aligned with an opening in the canopy). For this reason, we only used the average daily radiation relative to the top of canopy to compare with the model results. Although simulated days (LBA-MIP) include the entire range of daily averages of temperature, humidity, and net radiation observed during the ABRACOS campaign, the simulated period had significantly (t-test at $95\%$ confidence) warmer average temperatures at MDK ($0.67°$C) and higher net radiation at both RJA ($13.7\,\mathrm{W\,m^{-2}}$) and MDK ($18.5\,\mathrm{W\,m^{-2}}$) (Fig. S1). Therefore, we also compared the predicted light extinction profile with observations for overcast hours, when the direct radiation (based on the Weiss and Norman (1985) model) would be zero. Under overcast conditions, the model predictions of relative light extinction do not depend on the total incoming radiation (Longo et al., 2019). To evaluate the model's ability to represent changes in light environment throughout the canopy, we compare the light profiles of model and observations as a function of the cumulative tree area index (TAI). TAI is defined as the sum of leaf area index (LAI) and the branch wood area index (WAI). TAI profiles were estimated from published data at RJA (Simon et al., 2005) and near MDK (McWilliam et al., 1993).

## 2.2 Evaluation of long-term dynamics

To evaluate the model's ability to represent the long-term dynamics, we carried out multiple simulations intended to test the model's ability to describe regional variability as well as the structural and functional diversity of ecosystems in tropical South America.

First, to assess the model's ability to represent the distribution of biomass and leaf area index across tropical South America ($83°$W–$33°$W; $18°$S–$13°$N), we carried out a $1–°$ resolution, 500-year-long regional simulation, starting from near-bare conditions (1400-1900) to produce the simulated potential vegetation. In these simulations, we assumed a constant treefall disturbance rate of $1.4\%\,\mathrm{yr^{-1}}$ (Moorcroft et al., 2001), and allowed the occurrence of fire, using the original ED-1 implementation of fire disturbance model (Moorcroft et al., 2001). We then resumed the simulation in 1900, applying anthropogenic disturbance and ran the model until 2002, using a combination of land use transition matrices from Hurtt et al. (2006), nudged to match the initial conditions from Soares-Filho et al. (2006) in the Amazon. We specified the soil texture for each grid cell using data from Quesada et al. (2011) for the Amazon, RADAMBRASIL (de Negreiros et al., 2009) for non-Amazonian areas of Brazil, and IGBP (Tempel et al., 1996) for non-Amazonian areas elsewhere. Soil texture characteristics (sand, silt, and clay content) are used to determine the pedotransfer function, hydraulic and thermal conductivities, and heat capacity of soils (Longo et al., 2019). For the meteorological forcing, we used the Princeton University Global Meteorological Forcing Dataset (PGMF, Sheffield et al., 2006) for 1969 to 2008, which was recycled multiple times to simulate a period equivalent to 1400 through 2002.

We examined the relationship between regional above-ground biomass (AGB) and light and water availability by comparing the model's AGB predictions against regional observations of these two quantities. Specifically for light availability, we compared the model predictions against two estimates of average annual shortwave radiation: one calculated from the PGMF shortwave measurements and that were used to drive the model simulations and second the annual average downwelling shortwave

irradiance from the Clouds and the Earth's Radiant Energy System's Energy Balanced And Filled product (CERES-EBAF; Kato et al., 2013) between 2001 and 2017. Similarly, for rainfall, we compared the model's AGB prediction to annual rainfall estimates calculated from both the PGMF meteorological dataset and the average of annual precipitation calculated from the

145 Tropical Rainfall Measurement Mission's (TRMM) Multi-Satellite Precipitation Analysis Product (TMPA-3B43 V7; TRMM, 2011; Liu et al., 2012) between 1998 and 2017. For maximum cumulative water deficit (MCWD, mm), we assumed a constant monthly evapotranspiration ($ET_0 = 100$ mm) and monthly precipitation ($P$, mm) from TMPA-3B43, following previous studies for the Amazon region (Malhi et al., 2009a; Lewis et al., 2011; Aragão et al., 2018). For any month $t$, we defined the cumulative water deficit (CWD, mm) to be:

$$CWD(t) = \min\{1200, \max[0, CWD(t - \Delta t) + ET_0 - P(t)]\}, \tag{1}$$

where $\Delta t = 1$ month. This integration is continuously carried over the entire time series. For each calendar year, we defined MCWD as the maximum monthly value of CWD. Note that this estimate of MCWD assumes evapotranspiration values that are typical of tropical rainforests (i.e. $ET_0 = 100$ mm) and therefore should be regarded as a forest-equivalent MCWD, not an actual measurement. High values of MCWD indicate regions where water losses would be too high to maintain forests. This

baseline evapotranspiration ($ET_0$) is high for arid regions, and consequently we imposed a cap of $1200$ mm on CWD to avoid extremely high deficits at the most arid regions, where precipitation is insufficient to bring the water deficit back to zero.

We re-projected the estimates of shortwave irradiance, precipitation, and MCWD to the same grid as ED-2.2 using spatial averaging. For each environmental variable, we divided the grid cells into 20 quantile-based groups ($0 - 0.05$; $0.05 - 0.10$; ...; $0.95 - 1$), and obtained the average and the $90\%$ quantile range within each bin for the ED-2.2 model and three remote-sensing

estimates of aboveground biomass: Saatchi et al. (2011), Baccini et al. (2012), and Avitabile et al. (2016). We also compared the model's predictions of leaf area index (LAI) with estimates from the Moderate Resolution Imaging Spectroradiometer (MODIS, product MCD15A2H, Collection 6) (Yan et al., 2016). We used all cloud-free, high-quality data from MODIS-MCD15A2H that were available between August 2002 and July 2004. We selected this period to reduce temporal — and land-use related — differences between the model simulation and MODIS-MCD15A2H, and we aggregated the average LAI to $1°$ resolution

to be consistent with the ED-2.2 simulations. Finally, to evaluate the model's predictions of fire regime across tropical South America, we compared the average fire disturbance rate over predicted by ED-2.2 over one full cycle of the meteorological drivers (last 40 years of the simulation) with the Global Fire Emissions Database, version 4.1 (GFED4.1, Giglio et al., 2013; Randerson et al., 2018). Similarly to aboveground biomass and LAI, we aggregated GFED4.1 relative burned area to $1°$; because GFED4.1 data has a short overlapping period with the PGMF drivers, we used all available years (1997–2015) to

compute the average fire disturbance rate.

To evaluate the model's ability to predict emergent properties, we used published values of biomass and mortality obtained from the RAINFOR field inventory network in the Amazon (Phillips et al., 2004; Baker et al., 2004a, b) and the results from long-term simulations near the field inventory sites (Levine et al., 2016). Specifically, we investigated the relationship between aboveground biomass and long-term average mortality rates, and community averaged wood density across these sites. In

ED-2.2, typical wood density and background mortality rates are prescribed for each cohort, based on their plant functional

type (PFT). However, the model allows for coexistence of PFTs that can be modulated by abiotic and biotic factors such as soil texture, climate, and direct competition for limiting resources (e.g. light and water) between PFTs (Moorcroft et al., 2001; Longo et al., 2019). Consequently, mortality rates and community-level wood density are not prescribed parameters; instead, these properties emerge from the population structure comprised by different plant functional types (PFTs).

Finally, we tested both the forest structure predictions and the model's ability to represent variability of demographic rates at the two study sites (GYF and TNF) for which we had long-term forest inventory data. We ran the model from near-bare ground conditions for 500 years for each location, driven by the site-specific meteorological conditions, to obtain the near-steady state of each forest and to evaluate the forest structure predicted by the model at steady state. Several factors that are not well constrained for these sites may affect the simulations. For example, at both sites the meteorological drivers were not available for the period before the first inventory (in the case of TNF, meteorological data became available only two years after the first inventory). In addition, soil texture is known to be heterogeneous at local scale (e.g. Epron et al., 2006). Likewise, leaf phenology strategies within the local community can be diverse and heterogeneous (Bonal et al., 2000), and not known for the species included in the forest inventories. ED-2.2 model simulations are known to be modulated by such factors (e.g. Longo et al., 2018); therefore, to evaluate the sensitivity of mortality and the growth rates to these factors, we carried out an ensemble of simulations using all 48 combinations of (1) different initial times, between 5 and 60 years before the first inventory (intervals of 5 years); (2) soil texture, using clayey loam and sandy clay loam for GYF, and clayey sand and clay for TNF following the typical soils found at each site (Nepstad et al., 2002; Bonal et al., 2008); (3) assuming that trees were all evergreen or all drought deciduous. Only cohorts with DBH $\geq 10$cm were considered for the comparison of structure and demographic rates, because this was the minimum size of trees measured at both locations (Pyle et al., 2008; Rowland et al., 2014).

## 3 Results

### 3.1 Evaluation of energy and water fluxes

Both GYF and TNF had measurements of outgoing shortwave radiation ($\dot{Q}_{\mathrm{SW}}^{\uparrow}$) and outgoing photosynthetically active radiation ($\dot{Q}_{\mathrm{PAR}}^{\uparrow}$). Here, we compared the outgoing radiation instead of net radiation because net radiation includes incoming radiation, which are input variables for the model. The seasonal variation of both $\dot{Q}_{\mathrm{SW}}^{\uparrow}$ and $\dot{Q}_{\mathrm{PAR}}^{\uparrow}$ is close to observations at both sites (Fig. 1a-d). This result is expected because the seasonality of outgoing radiation is strongly modulated by incoming radiation. Nevertheless, the differences between simulated and observed $\dot{Q}_{\mathrm{SW}}^{\uparrow}$ and $\dot{Q}_{\mathrm{PAR}}^{\uparrow}$ are also small: the bias in the annual average of $\dot{Q}_{\mathrm{SW}}^{\uparrow}$ was $3.0\,\mathrm{W\,m^{-2}}$ (GYF) and $0.6\,\mathrm{W\,m^{-2}}$ (TNF), and the annual average bias of $\dot{Q}_{\mathrm{PAR}}^{\uparrow}$ was $-0.9\,\mathrm{\mu mol\,m^{-2}\,s^{-1}}$ (GYF) and $-1.6\,\mathrm{\mu mol\,m^{-2}\,s^{-1}}$ (TNF) (Fig. 1a-d). Likewise, the vertical profiles of daily averages of relative incoming PAR are close to the simulations, especially considering the large uncertainties on the observed tree area index (TAI) profile and that the simulated years are not the same as the observations (Fig. 2a,b). When we only consider overcast hours (less sensitive to local horizontal heterogeneities), the model shows even better agreement with model predictions, particularly at RJA, where

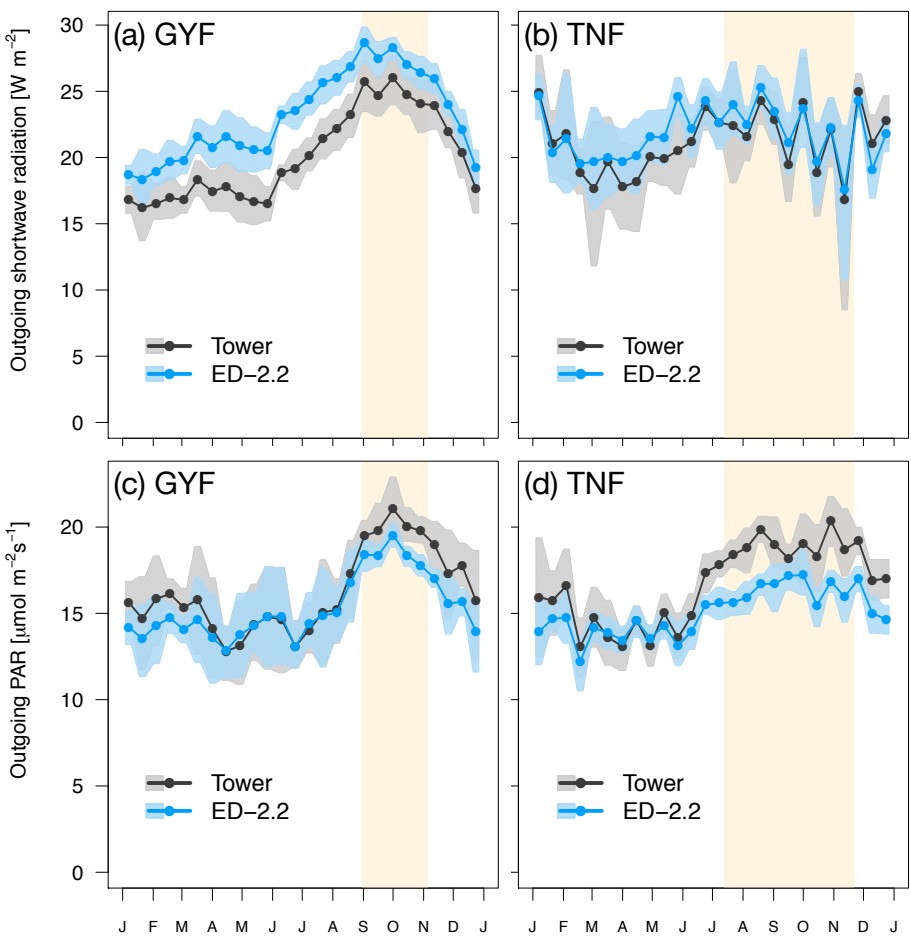

**Figure 1.** Mean annual cycle from fortnightly means of (a,b) outgoing shortwave radiation and (c,d) outgoing photosynthetically active radiation for (a,c) GYF and (b,d) TNF. Bands are the $95\%$ confidence interval of means, and rectangles in the background correspond to the site's climatological dry season.

the average profile was measured at the same site as the original observations, albeit at different years. While limited by the observation constrains, this comparison suggests that the model is able to reasonably reproduce typical light profiles.

Compared to tower observations, the model predicted higher mean sensible heat flux at both sites. At GYF, ED-2.2 captures the seasonal cycle very similarly to the observations, however, the fluxes are on average $19.5\,\mathrm{W\,m^{-2}}$ higher in ED-2.2 than measured by the tower (Fig. 3a). On the other hand, the model predicts significantly higher values at TNF (average bias $35.9\,\mathrm{W\,m^{-2}}$), in particular during the dry season (average dry-season bias $44.5\,\mathrm{W\,m^{-2}}$). The predicted seasonal cycle at TNF is very similar to the model prediction for GYF, whereas the tower measurements at TNF suggest much less variation between

wet and dry seasons (Fig. 3b). From the distribution of fluxes by time of the day, we found that ED-2.2 predicted higher than observed sensible fluxes during the afternoon at both sites (Fig. 3c,d). In addition, ED-2.2 was unable to capture events of

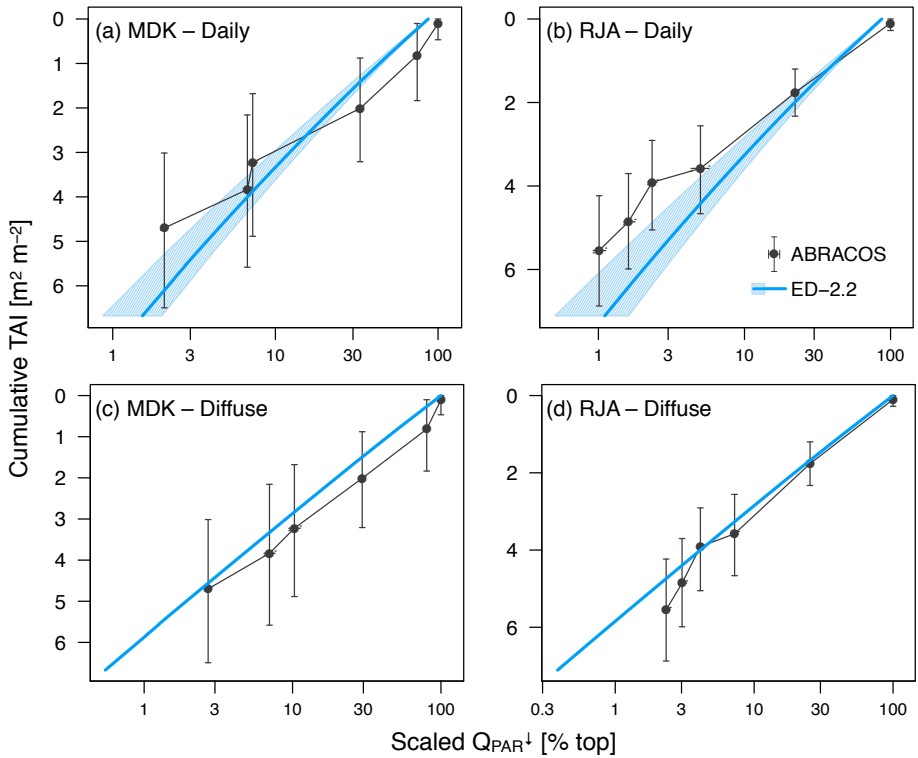

**Figure 2.** Comparison of downward photosynthetically active radiation profile relative to the top of the canopy. (a,b) Average of daily radiation as a function of cumulative tree area index (TAI); (c,d) average of times with overcast conditions (diffuse radiation only) for (a,c) Adolpho Ducke (MDK) and (b,d) Jaru (RJA). Shaded areas in the model correspond to the $95\%$ range of daily averages. Simulated subsamples for diffuse radiation only do not show variability because all plant functional types are assumed to have the same optical properties. Horizontal whiskers in observed values also correspond to the $95\%$ confidence interval of daily and diffuse subsamples, whereas the vertical whiskers correspond to the $95\%$ confidence interval of cumulative TAI at the points of measurement.

strong nighttime negative flux as measured at both towers. The predicted canopy air space temperature at GYF is typically $0.5\,^\circ\mathrm{C}$ lower than tower estimates (Fig. S2a). At TNF, the predicted canopy air space temperate shows close agreement with tower estimates during the wet season, and ED-2.2 predicts temperatures about $0.5\,^\circ\mathrm{C}$ higher than tower estimates (Fig. S2b).

Comparison with outgoing thermal infrared radiation, which is correlated with the vegetated surface temperature, show that the model typically predicts higher-than-observed values at daily scale (Fig. S3a,b), as a result of the model predicting higher fluxes during the afternoon (Fig. S3c,d). Therefore, the results suggest that the excessive sensible heat flux may be attributable to the model's overestimation of canopy temperature.

Water flux above canopy was well represented at both study sites (Fig. 4). The bias on annual average was $-0.23\,\mathrm{kg_W\,m^{-2}\,day^{-1}}$

for GYF and $+0.28\,\mathrm{kg_W\,m^{-2}\,day^{-1}}$ for TNF. The agreement was even better during the dry season (bias $-0.12\,\mathrm{kg_W\,m^{-2}\,day^{-1}}$ for GYF and $+0.12\,\mathrm{kg_W\,m^{-2}\,day^{-1}}$ for TNF). ED-2.2 correctly represents increased water flux during the dry season in TNF

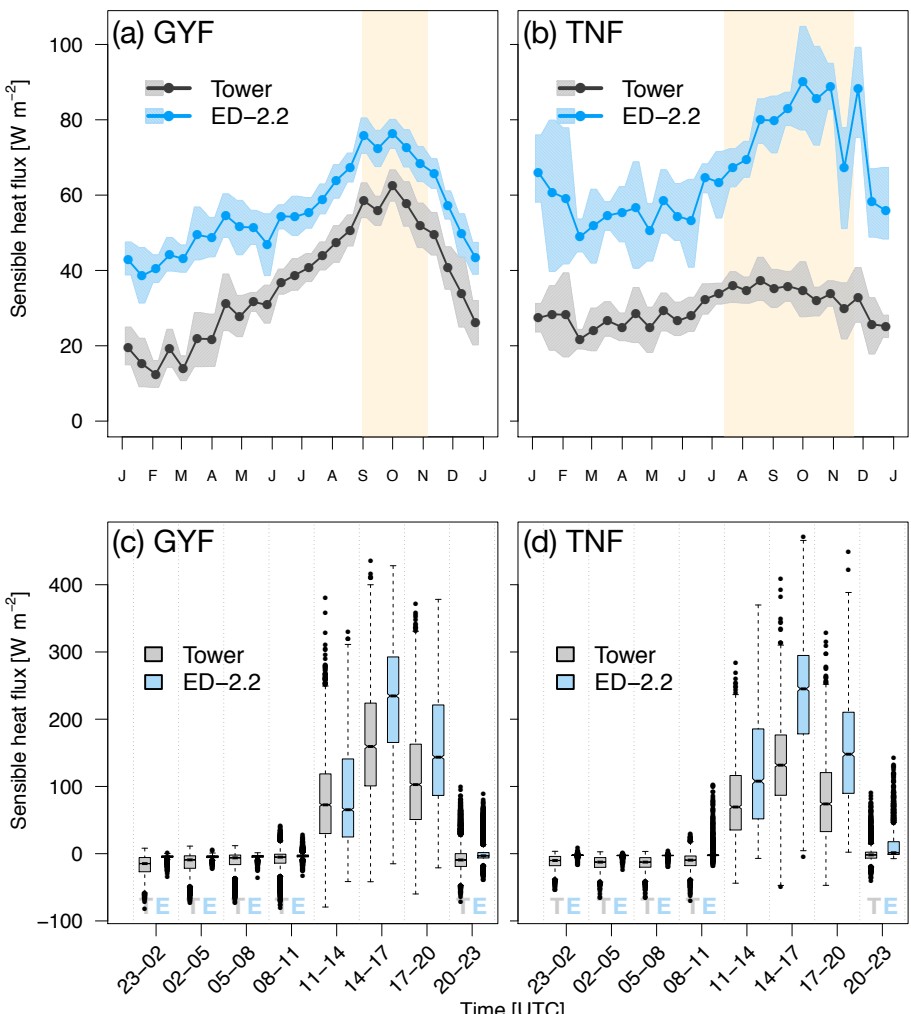

**Figure 3.** Mean annual cycle from fortnightly means of sensible heat flux for (a) GYF and (b) TNF. Bands are the 95% confidence interval of means, and rectangles in the background correspond to the site's climatological dry season. Box plot of sensible heat fluxes aggregated by time of day for all hours with available data for (c) GYF and (d) TNF.

(Fig. 4b), and that water flux does not increase during the dry season in GYF (Fig. 4a), although the model underestimates the fluxes during the first wet season (Dec-Feb) in GYF. Likewise, the diurnal cycle of water flux is well characterized at both sites, with median values predicted by the model similar to observations both during the day and during the night (Fig. 4c,d).
The mean canopy air space humidity at GYF shows excellent agreement during the wet season and slight underestimation during the dry season (Fig. S2d). In contrast, the model predictions at TNF show slightly lower values during the wet season, consistent with higher water fluxes, and agrees well during the dry season (Fig. S2d).

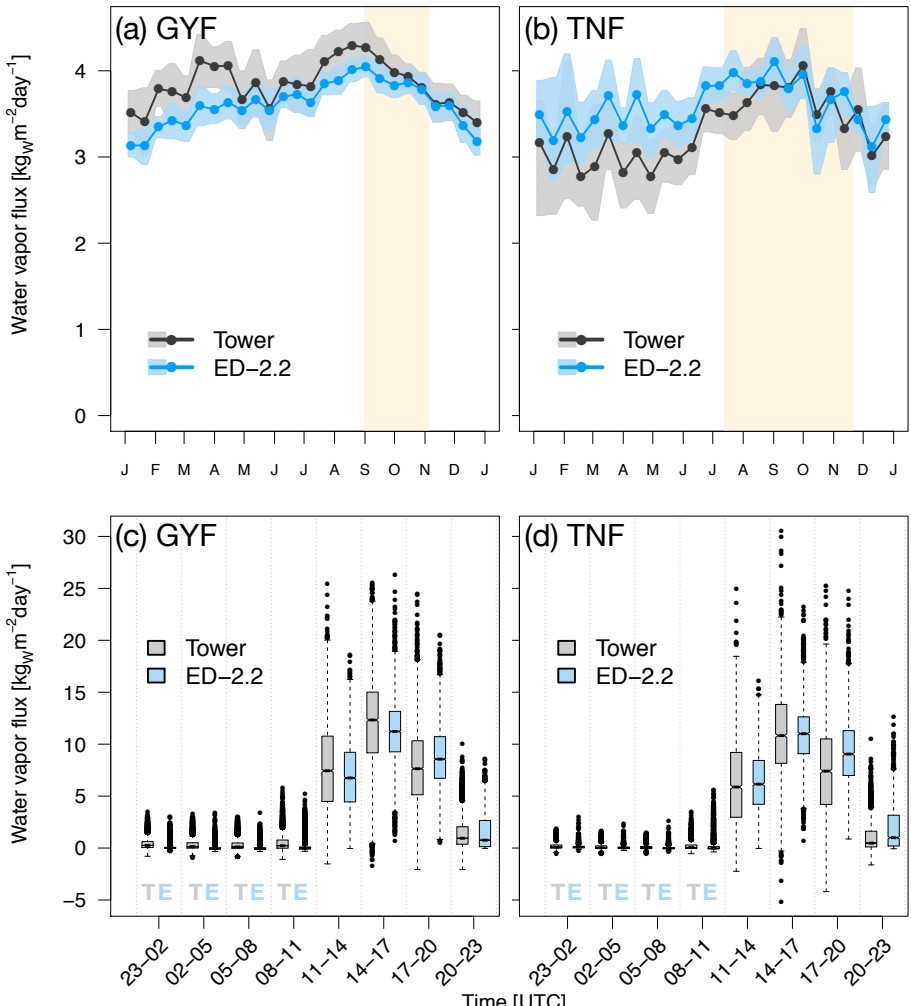

**Figure 4.** Mean annual cycle from fortnightly means of water flux for (a) GYF and (b) TNF. Bands are the 95% confidence interval of means, and rectangles in the background correspond to the site's climatological dry season. Box plot of water fluxes aggregated by time of day for all hours with available data for (c) GYF and (d) TNF.

## 3.2 Evaluation of productivity and respiration

When we compared gross primary productivity (GPP) with estimates from eddy covariance towers, the model captured the
235 weak seasonality with slightly higher values during the dry season at GYF although the magnitude was consistently lower (Fig. 5a), whereas it captured the magnitude of GPP at TNF but it could not represent the decreased GPP early in the dry season (Fig. 5b).

Ecosystem respiration from ED-2.2 is more seasonal than the tower estimates, and the model predicts the highest respiration rates during the wet season at both sites, whereas tower-based estimates do not show a consistent seasonal pattern. Based on

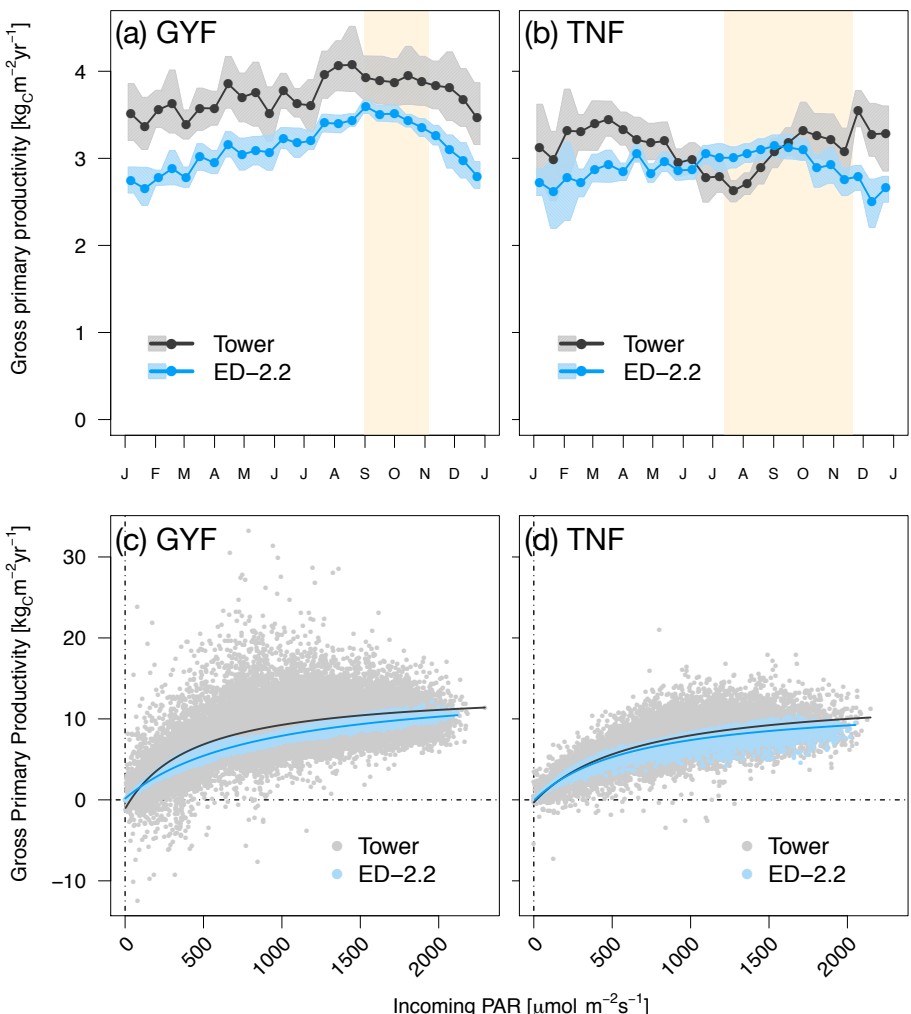

**Figure 5.** Mean annual cycle from fortnightly means of GPP for (a) GYF and (b) TNF. Bands are the $95\%$ confidence interval of means, and rectangles in the background correspond to the site's climatological dry season. Hourly averages of GPP as a function of incoming PAR for both model and tower based estimates at (c) GYF and (d) TNF.

tower estimates, respiration reaches the maximum early in the dry season in GYF (Fig. 6a), whereas the maximum occurs during the wet season in TNF (Fig. 6b). At both sites the model response is largely driven by fluctuations in soil moisture affecting heterotrophic respiration (Fig. 6c,d).

Ecosystem respiration is not directly measured from eddy covariance tower. The magnitude of ecosystem respiration from towers often depend on the choice of the $u^{\star}$ filter (Hayek et al., 2018b), and can be biased because the extrapolation of respiration from nighttime to daytime does not account the light inhibition of daytime leaf respiration, known as the Kok effect (Wehr et al., 2016). Therefore, we also compared the model predictions against results from other studies that focused on

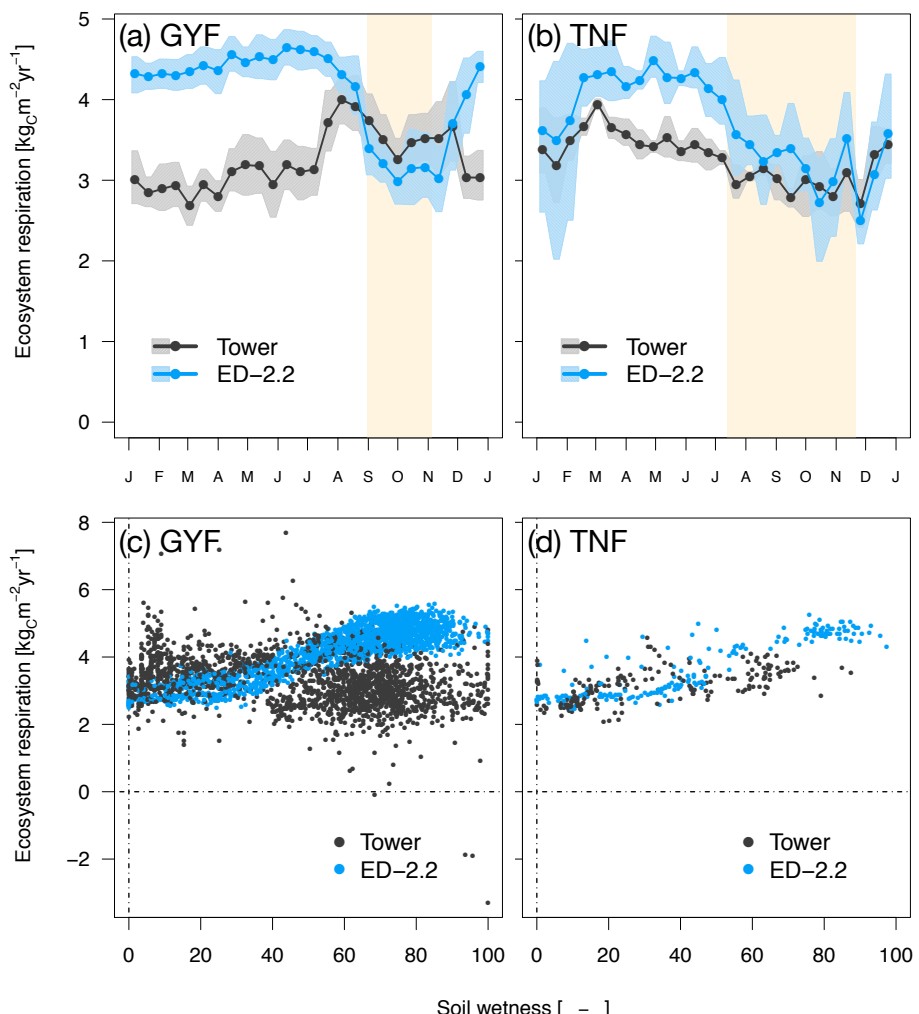

**Figure 6.** Mean annual cycle from fortnightly means of ecosystem respiration for (a) GYF and (b) TNF. Bands are the $95\%$ confidence interval of means, and rectangles in the background correspond to the site's climatological dry season. Daily mean ecosystem respiration for (c) GYF and (d) TNF as a function of daily mean relative soil moisture at (c) $20\,\mathrm{cm}$ and (d) $50\,\mathrm{cm}$, for days with both soil moisture measurements and tower-based estimates of ecosystem respiration.

measuring or estimating each component of the total respiration. For TNF, we used the respiration components estimated by a bottom-up assessment of the carbon cycle (Malhi et al., 2009b). For GYF, we estimated the respiration terms using previously published results and aggregated the components following Malhi et al. (2009b) (Supplement S1). The mean ecosystem respiration predicted by ED-2.2 overlaps with the $95\%$ confidence interval at both sites (Table 1). In GYF, differences in heterotrophic respiration are the largest and explain most of the difference in the total respiration, whereas in TNF the positive bias in total respiration is due to autotrophic respiration. Within the autotrophic respiration, ED-2.2 predicts stem respiration that is $89 - 134\%$ higher than the bottom-up assessment. In GYF the excessive stem respiration is compensated by leaf and

**Table 1.** Comparison of respiration between ED-2.2 and published values for GYF and TNF ($\overline{X} \pm \mathrm{SE}(\overline{X})$ for any variable $X$), using a bottom-up approach with published values. Values in parentheses are the standard error of the mean. For ED-2.2 the standard error was estimated from bootstrapping the annual means.

| | Paracou (GYF)[a] | | Tapajós (TNF)[b] | |
|---|---|---|---|---|
| | Bottom-up estimates | ED-2.2 | Bottom-up estimates | ED-2.2 |
| Ecosystem | $3.8 \pm 0.5$ | $4.03 \pm 0.12$ | $3.0 \pm 0.4$ | $3.80 \pm 0.10$ |
| Heterotrophic | $1.10 \pm 0.20$ | $1.66 \pm 0.03$ | $1.49 \pm 0.14$ | $1.432 \pm 0.024$ |
| Autotrophic | $2.8 \pm 0.5$ | $2.37 \pm 0.10$ | $1.5 \pm 0.4$ | $2.36 \pm 0.09$ |
| Soil+CWD[c] | $1.84 \pm 0.23$ | $2.20 \pm 0.05$ | $1.65 \pm 0.13$ | $1.97 \pm 0.04$ |
| Leaf | $1.4 \pm 0.4$ | $0.896 \pm 0.004$ | $0.7 \pm 0.4$ | $0.937 \pm 0.005$ |
| Stem | $0.504 \pm 0.019$ | $0.94 \pm 0.07$ | $0.38 \pm 0.10$ | $0.89 \pm 0.06$ |
| Root | $0.67 \pm 0.20$ | $0.542 \pm 0.030$ | $0.37 \pm 0.08$ | $0.534 \pm 0.026$ |

[a] Observed values for GYF are summarized in Supplement S1. Estimates were based on the approach described by Malhi et al. (2009b, c).

[b] Observed values for TNF are from Malhi et al. (2009b, c) and references therein.

[c] ED-2.2 does not have a separate coarse woody debris pool, therefore we compared the sum of both.

root respiration that are $36\%$ and $19\%$ lower than field estimates, respectively. As a result, the total autotrophic respiration at GYF is within one standard error from the expected rate based on the bottom-up analysis. In TNF, the reference leaf and root respiration are nearly half the magnitude for GYF; consequently, ED-2.2 autotrophic respiration is $57\%$ higher than the estimates by Malhi et al. (2009b).

### 3.3 Regional patterns of biomass

The model correctly predicts the extension of the Amazon forest (Fig. 7a), and it also represents the regional distribution of aboveground biomass within the Amazon biome compared to regional biomass maps from Saatchi et al. (2011), Baccini et al. (2012) and Avitabile et al. (2016), the latter being based on the other two maps. For example, ED-2.2 predicts higher aboveground biomass in the Guiana Shield, similar to estimates from Saatchi et al. (2011) and Avitabile et al. (2016) (Fig. 7b,d); the higher biomass near the border between Brazil, Peru, and Colombia, similar to Baccini et al. (2012) and Avitabile et al. (2016) (Fig. 7c,d); and the low biomass, open savanna area near the Brazil-Guyana-Venezuela border. The model generally predicts higher biomass than the three remote-sensing maps for most of the Amazon south of the Guiana Shield, particularly in the Western part of the Amazon (Fig. S4b-d), resulting in a peak in the distribution of biomass at $16.5\,\mathrm{kgC\,m^{-2}}$, whereas the high-biomass peaks ranged between $11.5 - 14.0\,\mathrm{kgC\,m^{-2}}$ for the remotely sensed estimates of aboveground biomass (Fig. 8). The ED-2.2 model and the remote-sensing estimates consistently predict relatively lower density function for intermediate values of biomass, and a pronounced peak of low biomass, even though the low-biomass peak predicted by ED-2.2 ($0.65\,\mathrm{kgC\,m^{-2}}$) is lower than the remote-sensing estimates ($1.0 - 3.4\,\mathrm{kgC, m^{-2}}$, Fig. 8). The shift in the low-biomass peak is mostly driven by ED-2.2 predictions of biomass in the savannas and xeric shrublands of Eastern Brazil, which were consistently lower than

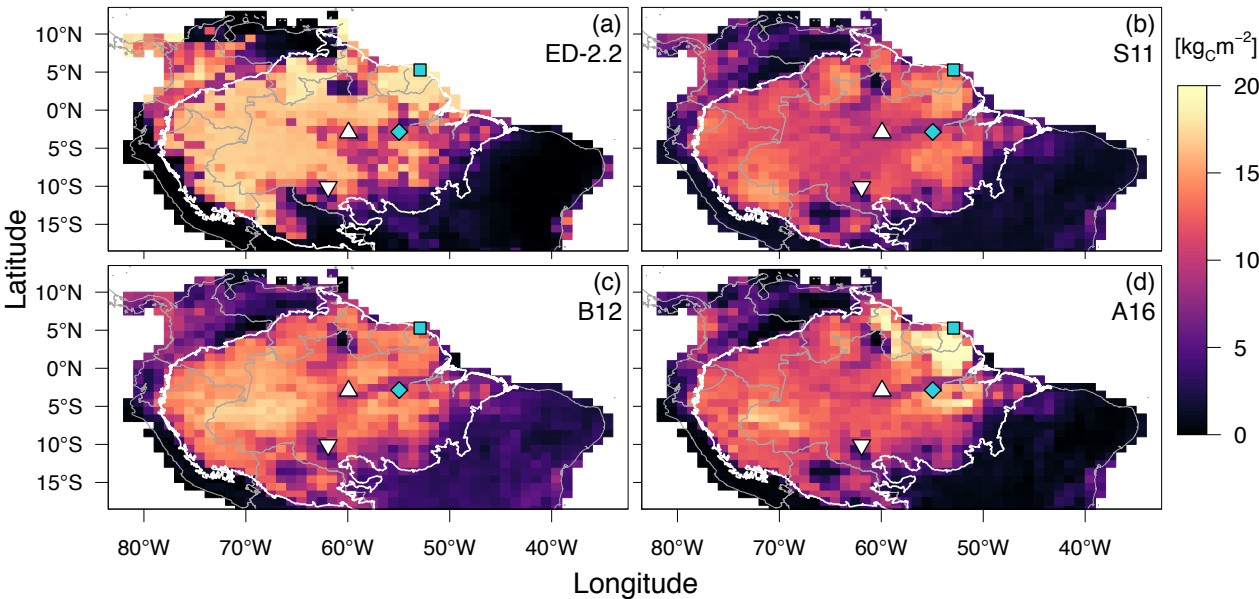

**Figure 7.** Comparison of aboveground biomass from (a) ED-2.2 and based on remote-sensing estimates from (b) Saatchi et al. (2011), (c) Baccini et al. (2012), and (d) Avitabile et al. (2016). Remote-sensing maps were aggregated to $1°$ resolution. Blue points represent the focus sites of Paracou (GYF, $\square$) and Tapajós (TNF, $\Diamond$). White points represent the sites used for radiation profile evaluation: Ducke (MDK, $\triangle$), and Jaru (RJA, $\triangledown$). White contour is the domain of the Amazon biome, and grey contours are the political borders. Maps of the differences between ED-2.2 and remote-sensing are shown in Fig. S4.

the remote-sensing estimates (Fig. S4). We also found that ED-2.2 predicted a similar extent of LAI over the Amazon region when compared to MODIS estimates (Fig. S5). However, ED-2.2 predicted lower LAI than MODIS for most of the Amazon, in particular along the arc of deforestation, and higher LAI in northwestern Colombia and Central Brazil (Fig. S5c).

The predicted transitions between the Amazon biome and the Cerrado (savanna biome in Central Brazil) and Los Llanos (grassland area in Venezuela) are driven by increased fire activity, consistent with the Global Fire Emission Database (GFED4.1 Giglio et al., 2013; Randerson et al., 2018) regional distribution of burned area ($\alpha$; Fig. 9). In addition, ED-2.2 correctly predicted higher burned area in the savanna region at the Brazil-Guyana-Venezuela border and low fire activity over most of the Amazon biome and in the Caatinga (low-biomass, semi-arid region in Northeastern Brazil) (Fig. 9). In contrast, ED-

2.2 underestimated fire activity in El Beni (Bolivian savannas, $\Delta\alpha = \alpha_{ED-2.2} - \alpha_{GFED4.1} = -6.8\%\,yr^{-1}$) in the Colombian Los Llanos ($\Delta\alpha = -9.2\%\,yr^{-1}$), while it overestimated fire disturbance in forest areas near the border of Brazil and Bolivia ($\Delta\alpha = 1.9\%\,yr^{-1}$), the coastal areas of Brazilian states of Maranhão and Piauí, immediately east of the Amazon biome ($\Delta\alpha = 5.0\%\,yr^{-1}$), and areas in Eastern Brazil south of $15°S$ ($\Delta\alpha = 3.7\%\,yr^{-1}$) (Fig. 9; S6). Because the original fire model in ED-2.2 does not predict fire ignitions based on human activities (Moorcroft et al., 2001; Longo et al., 2019) ED-2.2 does not

predict the burned areas along the arc of deforestation in the Southern and Eastern edges of the Amazon biome (Fig. 9a).

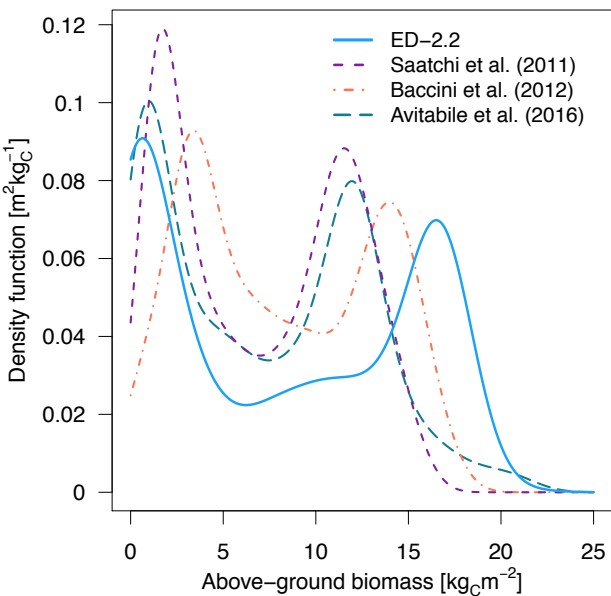

**Figure 8.** Density functions of aboveground biomass predicted by ED-2.2 and based on remote-sensing estimates from Saatchi et al. (2011), Baccini et al. (2012), and Avitabile et al. (2016); points used to derive the distribution are the same as Fig. 7, and the density functions were calculated using the same bin width across the range of biomass from the model and the remote-sensing based maps.

The predicted spatial variability of total carbon stocks in the region also emerged from variation in the environmental conditions such as variability in available light and water (Fig. 10;S7). Both ED-2.2 and the three remote-sensing estimates of biomass consistently showed the highest average biomass ($11.7 - 15.1 \, \mathrm{kgC \, m^{-2}}$) at $195 \, \mathrm{W \, m^{-2}}$, and the sharpest decline as a function of increased irradiance ($0.53 - 0.77 \, \mathrm{kg_C \, W^{-1}}$) near $225 \, \mathrm{W \, m^{-2}}$ (Fig. 10a; S7a). Similarly, the relationship between annual precipitation and above-ground biomass was consistent between model and remote-sensing estimates, with the highest changes in average biomass by increase in annual rainfall occurring between $1500$ and $2200 \, \mathrm{mm \, yr^{-1}}$, and relatively stable values of above-ground biomass above $2500 \, \mathrm{mm}$ (Fig. 10b). Some of the variability of simulated biomass at low ($< 1000 \, \mathrm{mm}$) annual rainfall can be attributed to discrepancies between TMPA-3B43 and PGMF, which was used to drive the simulations, as the simulated biomass is consistently low where PGMF estimates of annual rainfall are less than $1000 \, \mathrm{mm}$ (Fig. S7b). The increasing dry season severity, summarized by MCWD, has a strong association with decreasing average aboveground biomass in both the model and the remote-sensing estimates when the annual MCWD is less than $500 \, \mathrm{mm}$. The strongest declines in average biomass occurred at mean annual MCWD between $300 - 350 \, \mathrm{mm}$ in both ED-2.2 and the remote-sensing estimates (Fig. 10c; S7c). It must be noted, however, that the transition between high biomass and low biomass when MCWD $300 - 350 \, \mathrm{mm}$ is substantially more pronounced in ED-2.2 ($-0.075 \mathrm{kgC \, m^{-2} \, mm^{-1}}$) than in the remote-sensing estimates (between $-0.041$ and $-0.050 \mathrm{kgC \, m^{-2} \, mm^{-1}}$) (Fig. 10c).

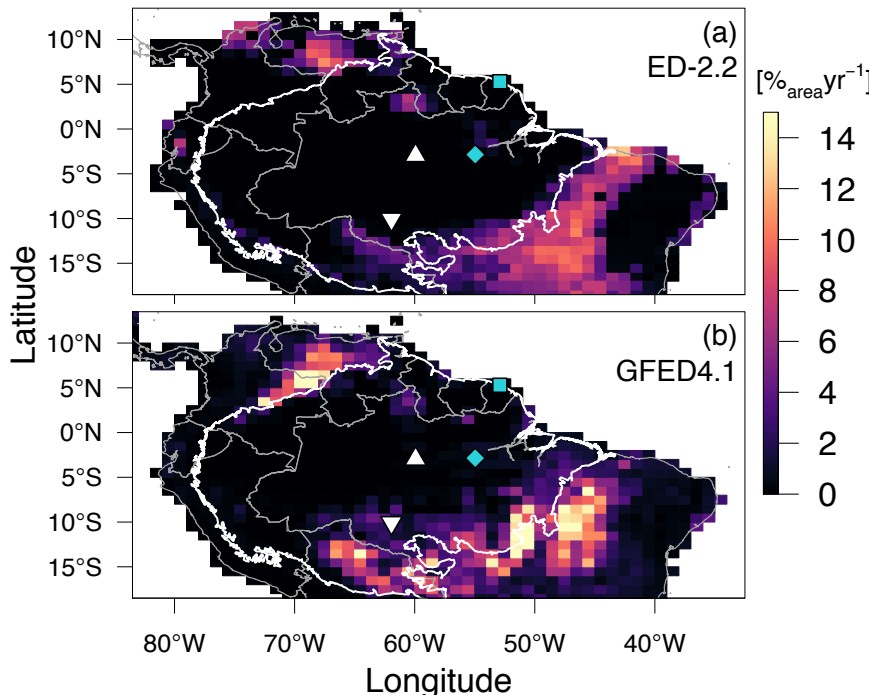

**Figure 9.** Comparison of average burned area (a) predicted by ED-2.2 for a 40-year average driven by the Princeton Global Meteorological Forcing (PGMF (1969–2008); Sheffield et al., 2006), and (b) obtained from the Global Fire Emission Database (GFED4.1 (1997–2015); Giglio et al., 2013; Randerson et al., 2018), aggregated to 1° resolution. The location of focus sites of Paracou (GYF, □) and Tapajós (TNF, ◇), and the sites used for radiation profile evaluation: Ducke (MDK, △), and Jaru (RJA, ▽) are shown for reference. Thick contour is the domain of the Amazon biome, and thin contours are the political borders. The map of the difference between ED-2.2 and GFED is shown in Fig. S6.

### 3.4 Assessment of forest function and structure

In addition to the total carbon stocks, the regional variability in forest function and composition at steady state is generally well characterized by ED-2.2. The range and variability of biomass across the network are similar to the range and variability observed across most of the network (Fig. 11a). However, the model does not accurately predict the AGB of individual plots at the wettest sites (black dots in Fig. 11a). The model predicts lower-than-observed biomass at the drier sites located in Bolivia (red dots in Fig. 11a) because of frequent fires occurring in the model simulation. Also, both the model and the field measurements show similar negative correlation between biomass and mortality rates (Fig. 11b), and a similar positive correlation between biomass and the mean wood density (Fig. 11c), albeit significant differences exist in biomass for any given value of mortality or wood density (Fig. 11b,c). Both wood density and mortality rates are related to the abundance of pioneers or late successional individuals both in the model (Moorcroft et al., 2001) and in observations (e.g. Chave et al., 2009; Kraft et al., 2010), suggesting that the model characterizes the variability in forest composition within the Amazon region. Fewer sites

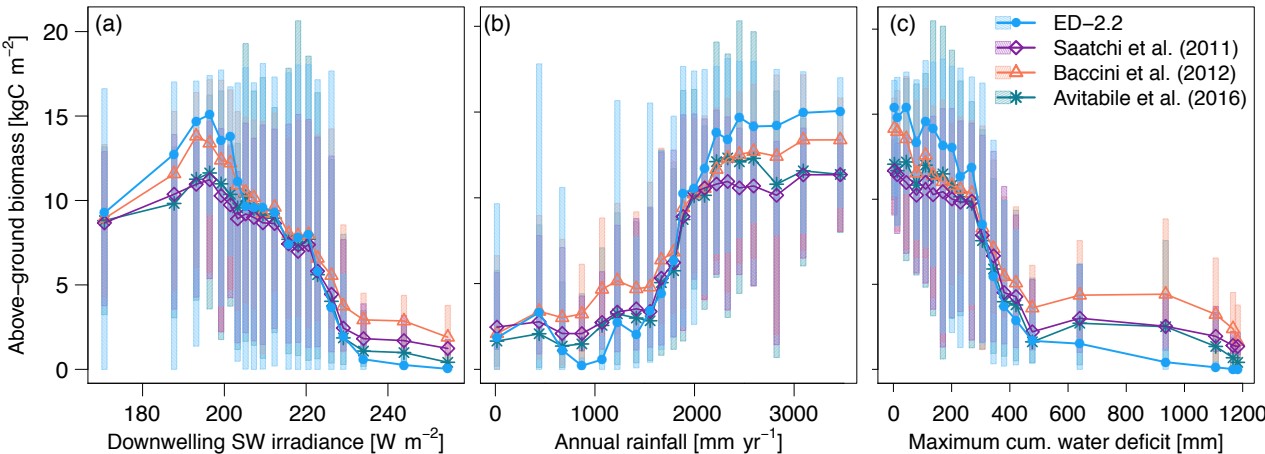

**Figure 10.** Average biomass predicted by ED-2.2 and based on remote-sensing maps from Saatchi et al. (2011), Baccini et al. (2012), and Avitabile et al. (2016), aggregated by annual averages of (a) downwelling shortwave irradiance; (b) mean annual precipitation; and (c) maximum cumulative water deficit. For each annual average of environmental properties, grid points were grouped into 20 quantile bins: points represent the average within each bin, and shaded area corresponds to the 90% quantile range within each bin. Data source for annual means: surface downwelling shortwave irradiance data (CERES EBAF-Surface Ed4.0; Kato et al., 2013); precipitation (TMPA-3B43; Liu et al., 2012); maximum cumulative water deficit was based on the same approach as Malhi et al. (2009a), using the TMPA-3B43 precipitation. The distribution of average biomass based on the meteorological drivers used by the ED-2.2 simulation (Sheffield et al., 2006) is shown in Fig. S7.

had estimates of both mortality rates and net primary productivity (NPP). At the available sites, the average predicted values at steady state ($1.29\,\mathrm{kgC\,m^{-2}\,yr^{-1}}$) were slightly lower than values reported in the literature ($1.35\,\mathrm{kgC\,m^{-2}\,yr^{-1}}$) (Fig. 11d). Consistent with previous observations in the Amazon (e.g. Quesada et al., 2012), the ED-2.2 predictions indicate a positive

correlation between NPP and mortality. However, this correlation is highly uncertain because of the limited number of plots ($n = 7$) and the large variability in both NPP and mortality across sites (Fig. 11d).

The model captured the general distribution slope of stem demographic density (abundance) at GYF (Fig. 12a), and both the total basal area and the typical distribution of basal area for individuals with DBH between $20$ and $50\,\mathrm{cm}$ (Fig. 12b). In contrast, the model predicted a lower contribution from trees with $\mathrm{DBH} < 20\,\mathrm{cm}$ to both abundance and basal area, and a

320 higher contribution of individuals with $\mathrm{DBH} > 50\,\mathrm{cm}$ to basal area (Fig. 12a,b). Model comparison with TNF data also showed that the model reproduced the main characteristics of the forest structure, although the slope of abundance as a function of size was steeper in the model compared to observations (Fig. 12c). Basal area structure, on the other hand, showed good agreement with field inventory data and total basal area was within $5\,\%$ of the observed basal area.

Total mortality rates were generally higher in ED-2.2 simulations than based on the observed rates at both GYF and TNF

(Fig. 13a,b), and the model showed little variability in mortality between different census intervals. Most of the modeled

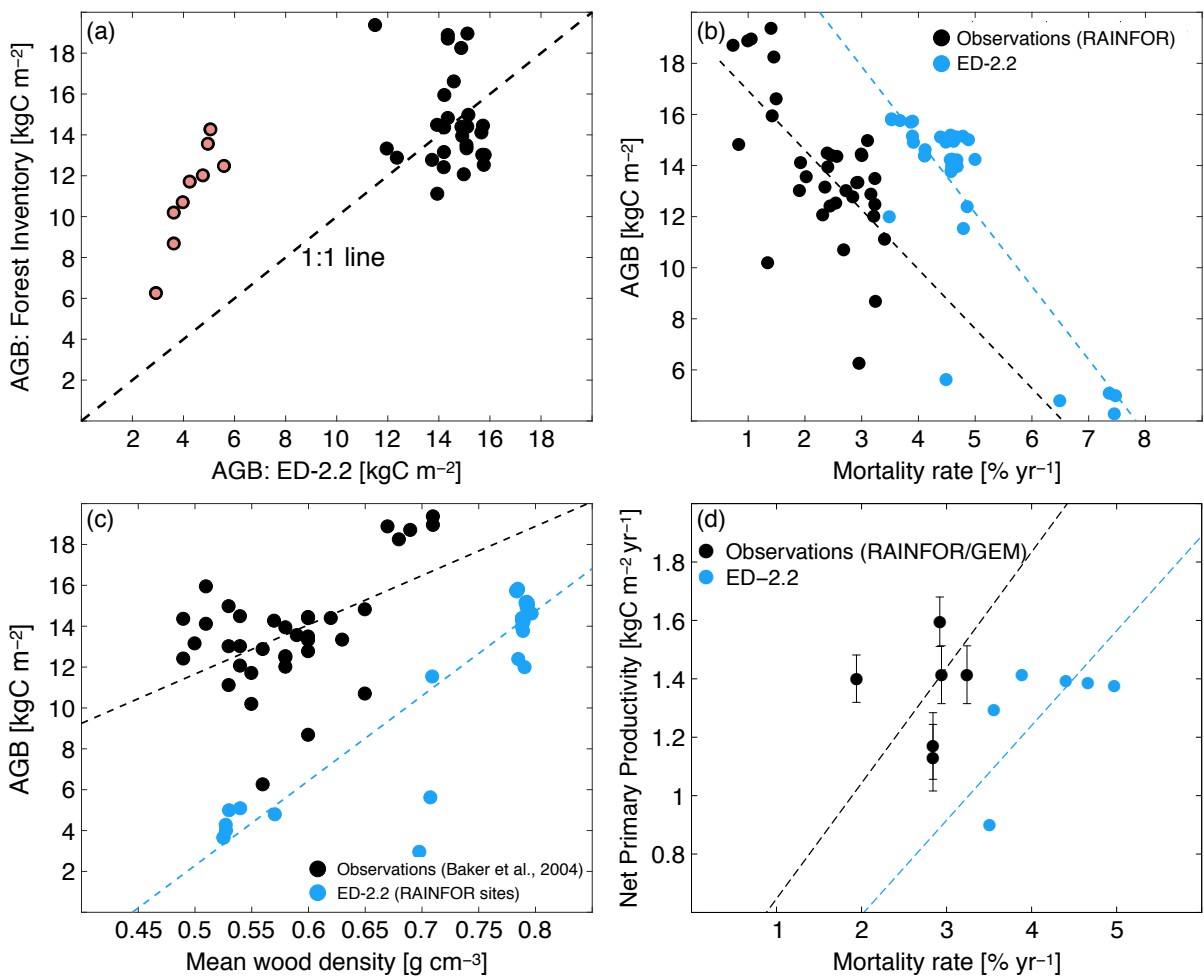

**Figure 11.** Evaluation of the ED-2.2 biomass and relationship between biomass and productivity and ecosystem function. (a) (Redrawn from Knox et al., 2015) Comparison of model predictions of mean aboveground biomass (AGB) with field estimates at multiple sites presented in Baker et al. (2004a, b). To be consistent with field measurements, model estimates included only biomass from living individuals with DBH $\geq 10$ cm (diameter at breast height). Points in red are from drier forest sites in Bolivia. (b,c) (Redrawn from Levine et al., 2016). Steady-state aboveground biomass distribution across the Amazon as a function of (b) mortality rates and (c) basal-area weighted mean wood density, for both the ED-2.2 model (blue circles) and based on field observations (black circles). Source of field-based estimates: Phillips et al. (2004); Baker et al. (2004a, b). (d) Steady-state net primary productivity (NPP) as a function of mortality rates, for both the ED-2.2 model (blue circles) and based on field observations (black circles; error bars are the reported standard error). Source of field-based estimates: mortality rates (Phillips et al., 2004, 2010); NPP (Aragão et al., 2009; Malhi et al., 2015).

mortality was due to background mortality, which is a combination of density-independent factors such as aging and treefall mortality, both assumed time-invariant in the model (Moorcroft et al., 2001). While the mortality due to environmental constrains (density-dependent) showed interannual variability, its magnitude was small, never exceeding $0.5\%_{\mathrm{AGB}}\,\mathrm{yr}^{-1}$. The high

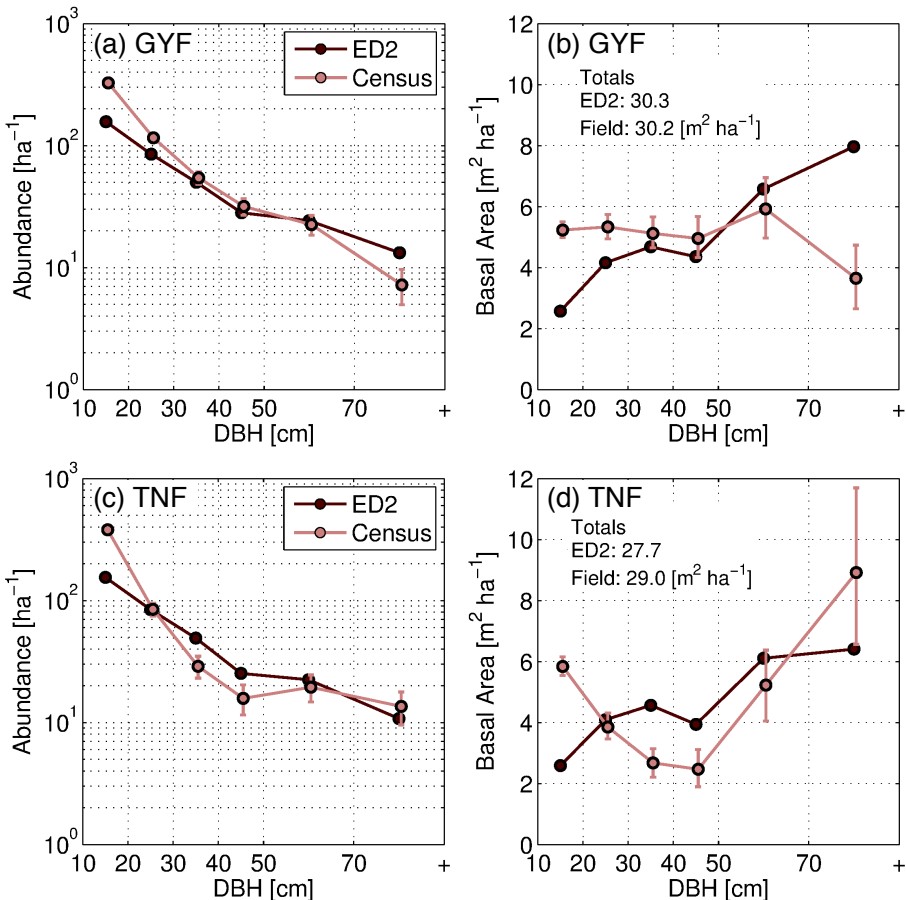

**Figure 12.** Evaluation of the ED-2.2 estimated size structure at the (a,b) GYF and (c,d) TNF sites, expressed in terms of abundance (a,c) and in terms of basal area (b,d). Confidence intervals on inventory analysis at 95% are shown with error bars. This is the confidence of the inventory sample representing the broader population (representativeness), as measurement error is assumed to be small. The confidence interval on abundance is estimated by a Poisson counting process. The confidence on the basal area is estimated by generating an ensemble of 100 random samples of the inventory, and quantile regression of the basal area sums of those re-samples. Total basal area predicted by ED-2.2 and observed in the census are shown in panels (b,d).

mortality rates in the model is mostly attributable to early-successional trees, for which the modeled mortality rates was near

$7.5\,\%_{\mathrm{AGB}}\,\mathrm{yr}^{-1}$, or 5-fold higher than late-successional, whereas observations were typically $3.0\,\%_{\mathrm{AGB}}\,\mathrm{yr}^{-1}$, or 3 times higher than late-successional. Likewise, growth rates were also higher than in observations, particularly at GYF (Fig. 13c), whereas the values were closer to observations for most of the simulated period at TNF (Fig. 13d). Growth rates of both sites are significantly different, which could be related to the particularly nutrient-poor soils at GYF (e.g, Baraloto et al., 2005), and that these ED-2.2 simulations did not account for nutrient limitation. Alternatively, higher growth rates may be due to tree allometry and

low allocation to living tissues and contributing to high accumulation rates on structural tissues.

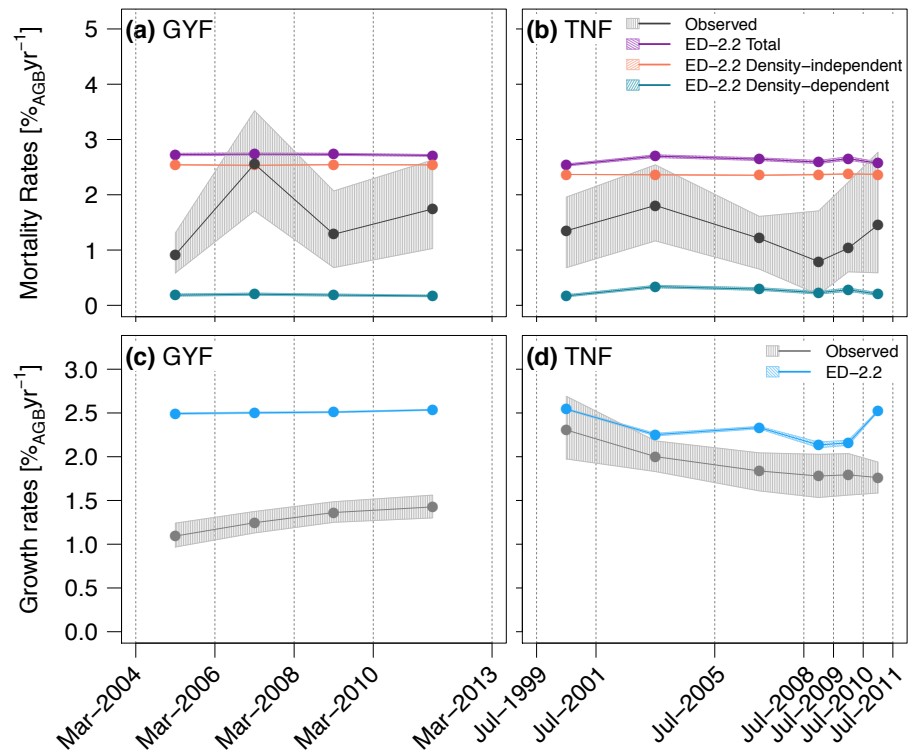

**Figure 13.** Comparison of (a,b) mortality rates and (c,d) growth rates obtained from simulations and forest inventories for the (a,c) GYF and (b,d) TNF sites (redrawn from Longo et al. (2018)). Vertical lines are the approximate times of forest inventory surveys, and bands associated with observations correspond to the 95% confidence interval, obtained from bootstrap (see Longo (2014) for further details), and bands associated with model results are the range of simulations with different soil texture, leaf phenology, and initial time. To be consistent with the field plot protocol, only those cohorts with $DBH \geq 10$ cm (diameter at breast height) were included in the model estimates.

## 4 Discussion

### 4.1 Water and energy fluxes

The comparisons with eddy covariance towers demonstrated the ability of ED-2.2 to simulate the magnitude and seasonality of both the evapotranspiration fluxes and the water storage in the canopy air space (Fig. 4;S2). The good agreement of evapotran-

spiration in the tropical sites using ED-2.2 contrasts with previous assessments using ED-2.1 for temperate sites, which found significant negative biases in simulated evapotranspiration and attributed the bias to the model overestimating the impacts of water stress on stomatal conductance (Matheny et al., 2014; Walker et al., 2014). The average ratio between canopy evaporation and total rainfall ranged from $7-11\%$ at the two forest sites tested here (Guyaflux, GYF; and Tapajós, TNF), which is at the low end but in the same range of values founds in previous studies ($9-20\%$; Tobón Marin et al., 2000, and references therein).

The hydrological cycle, on the other hand, showed some important deviations in absolute value, particularly near the surface,

despite being consistent with observation in relative terms (Fig. S9). Large biases in absolute soil moisture were caused by the mismatches in the soil hydraulic properties that ultimately define the residual moisture and wilting point, field capacity, and porosity, thence the range of possible values of soil moisture. Soil hydraulic properties were derived from texture characteristics previously published for all sites. In ED-2.2 these properties were simplified to a single fraction assumed constant at every patch and throughout the profile, whereas in reality soil properties are known to vary significantly within the same area and with depth (e.g. Epron et al., 2006). Moreover, soils in ED-2.2 are assumed to be mineral, whereas in reality macropores and soil organic content can substantially affect such properties (Saxton and Rawls, 2006; Fisher et al., 2008).

The model also realistically represented both the net absorption of visible irradiance and the vertical distribution throughout the canopy. The model showed similar magnitude and seasonality of the outgoing shortwave radiation, including photosynthetically active radiation, at the two long-term sites (Fig. 1), and the average light level profiles within the canopy (Fig. 2). Agreement was even higher during cloudy conditions (Fig. 2c,d), when the spatial distribution of individual trees has less of an impact on local variability in the light profile (Mercado et al., 2009). Differences were more significant for photosynthetically active radiation than for total shortwave radiation, particularly during the dry season (Fig. 1c,d). These differences may result from two factors. First, leaf optical properties in ED-2.2 are assumed constant for any given PFT, whereas observations indicate that leaf reflectivity depends on leaf age (Toomey et al., 2009; Chavana-Bryant et al., 2017). Second, ED-2.2 represents canopy structure in only one dimension for each patch, the effect of neighboring trees (or their absence) is not represented. A full three-dimensional approach similar to Morton et al. (2016) may not be feasible within the ED-2.2 because of the intensive computational burden and that the model does not represent the actual position of individual trees. Alternatively, the perfect plasticity approach (Purves et al., 2008; Farrior et al., 2013), in which finite-crown individual trees are arranged to maximize light access, has been recently adapted to another cohort-based model, the Functionally-Assembled Terrestrial Ecosystem Simulator (FATES, Fisher et al., 2018). The perfect plasticity approach has the advantage of allowing trees of similar size to experience the same light levels (as opposed to the sequential light interception in ED-2.2), which could contribute to improve the light extinction profile in closed canopy forests. Importantly, while the current representation of vertical light distribution in ED-2.2 may affect light availability of individual cohorts, it does not imply that the simulated understory in ED-2.2 is excessively dark. In fact, when we compared the modeled and observed vertical structure of diffuse light (which is less sensitive to the observed position of trees than direct light) we found that the model slightly overestimates understory light levels (Fig. 2).

The model predictions of canopy air space temperature at TNF generally matches observations well, except for a slight overestimation during the dry season at TNF (Fig. S2). However, the sensible heat flux also tends to be higher than tower estimates, particularly at the drier TNF site (Fig. 3). The better agreement of simulated water fluxes with observations relative to sensible heat flux is very common among land surface models (Best et al., 2015). One possible cause of this disagreement is that observations from eddy flux towers typically contain significant sources and sinks from lateral advection and air drainage that may result in departures from energy closure by as much as $30\%$ (Tóta et al., 2008; da Rocha et al., 2009; Leuning et al., 2012; Stoy et al., 2013). Energy conservation is an important requirement for terrestrial biosphere models, particularly in the context of coupled biosphere-atmosphere modelling studies where they need to ensure conservation of energy at the lower boundary of the atmosphere. In ED-2.2, conservation of energy is checked every model time step (Longo et al., 2019). Since ED-2.2

does not incorporate lateral advection of air within the plant canopy, in situations where significant lateral advective transport of energy occurs within the canopy, the model may compensate for the absence of lateral transport through an increased or decreased rate of vertical eddy fluxes. Haughton et al. (2016) suggested that parameterization problems, rather than energy conservation, are the most likely cause for biases in land surface models. One potential issue in this regard in ED-2.2 is that the heat capacity of branches and leaves could be biased, allowing greater variability of temperature and higher sensible heat fluxes at the expense of reduced storage. To our knowledge, no long-term measurements of leaf or branch temperature exist for the Amazon sites, but differences in outgoing thermal-infrared irradiance (Fig. S3) suggest that the observed vegetation temperature during the afternoon may be lower and less variable than the model predictions. Additional measurements of leaf and wood heat capacity for tropical forests could improve the accuracy of leaf and branch temperatures in the model.

## 4.2 Carbon fluxes and carbon storage

Comparisons of GPP between the ED-2.2 model and tower-based estimates show that the model captures both the magnitude of GPP and the typical GPP response to light (Fig. 5). In ED-2.2, the seasonality at tropical forest sites is mostly driven by light, thus the maximum productivity often occurs during the dry season (Fig. 5a,b). Estimates of GPP based on eddy covariance towers, however, suggest a more complex pattern, with minimum occurring either during the wet season (GYF) or the transition from wet to dry season (TNF), and modestly increasing productivity during the dry season (Fig. 5a,b; see also Bonal et al., 2008; Restrepo-Coupe et al., 2013, 2017). GPP depends on multiple processes, and thus differences between ED-2.2 and tower estimates may be due to several factors including biases in the seasonality of leaf area and photosynthetic properties of leaves. In particular, empirical studies have shown that the seasonality of GPP in Amazon forests is linked to seasonal variation of photosynthetic capacity and leaf phenology (Wu et al., 2016; Restrepo-Coupe et al., 2017), whereas in the current implementation of ED-2.2 both the leaf turnover rate and the maximum photosynthetic capacity are assumed, for simplicity, to be constant. Work from several modeling studies (Kim et al., 2012; De Weirdt et al., 2012; Medvigy et al., 2013), and a leaf-level carbon optimization model (Xu et al., 2017) have shown that incorporating such seasonality into the dynamics of leaf longevity and photosynthetic capacity significantly improves predictions of the seasonality of carbon fluxes in tropical forests. However, data from multiple sites may be needed to develop a generalizable phenology model for evergreen forests.

The model estimates of ecosystem respiration were generally higher and more seasonal than the expected values either from eddy covariance tower estimates or from a bottom-up assessment, especially at GYF (Fig. 6; Table 1). In the model, the seasonality of ecosystem respiration was nearly exclusively driven by the seasonality in heterotrophic respiration, with a significant decline in the dry season due to lower soil moisture (Fig. 6c,d), whereas eddy covariance tower estimates suggest a dry-season reduction in respiration only at TNF (Fig. 6a,b; Restrepo-Coupe et al., 2017; Aguilos et al., 2018). The ED-2.2 assumption that heterotrophic respiration at lower soil moisture (Longo et al., 2019) is consistent with observations for total respiration at GYF (e.g. Fig. 3 of Aguilos et al., 2018); however, the magnitude of the heterotrophic response to moisture is likely overestimated in the ED-2.2 model.

Total autotrophic respiration estimates from ED-2.2 were within range with independent bottom-up estimates for both sites, albeit only marginally within the $95\%$-confidence level (2 standard errors) of the bottom-up estimates at TNF (Table 1). At both

sites, the simulated autotrophic respiration was driven by leaf and stem respiration, which contributed with roughly the same proportion to autotrophic respiration. In contrast, the bottom-up estimates at both sites suggest leaf respiration was 2–3 fold higher than stem respiration (Table 1). One possibility is that the allocation of carbon gains to tissue growth was overestimated in ED-2, which is consistent with the overestimated growth rates compared to forest inventory plots (Fig. 13). It must be noted, however, that several terms estimated from observations also carry large uncertainties and assumptions. For example, stem respiration is typically measured near the surface (e.g Chambers et al., 2004; Stahl, 2010), which may introduce biases given that branches may have significantly higher respiration rates (Cavaleri et al., 2006). Furthermore, observed differences of expected values between sites are generally much larger than the differences obtained by the model, and such differences may reflect true differences of plant community functioning between sites, or differences in sampling and techniques (Malhi et al., 2009b). Improved measurements of the different terms of ecosystem respiration would allow for better constrains on the individual processes driving the total respiration.

In addition to tropical sites, the model's ability to represent productivity and respiration has been previously assessed for temperate ecosystems. The model showed excellent agreement in magnitude and seasonality of both net ecosystem productivity (NEP) and gross primary productivity (GPP) at three flux tower sites installed at Harvard Forest, in particular when the model was initialized with ground-based or remote-sensing-based data (Antonarakis et al., 2014). In contrast, the model showed significant biases in net primary productive under ambient $CO_2$ at two Free-air $CO_2$ Enrichment (FACE) sites in Southeastern United States, overestimating NPP at Duke (evergreen forest) and underestimating at Oak Ridge (deciduous forest) (Walker et al., 2014). Also, a previous model-intercomparison study for eleven North American tower sites also revealed significant negative biases in ED-2.1, although the model inter-annual variability of GPP and ecosystem respiration were within the observed range for both deciduous and evergreen sites (Keenan et al., 2012). In addition, a wavelet analysis of the normalized error across 9 tower sites across North America suggested that the errors in net ecosystem exchange (NEE) are dominated by sub-annual (but longer than daily) time scales (Dietze et al., 2011). Together, these results indicate the need of quantifying which processes and parameters contribute the most to model uncertainties in order to improve the model predictions using ED-2.2, which is currently being pursued (Fer et al., 2018; Raczka et al., 2018).

Finally, because ED-2.2 solves the carbon dioxide cycles at sub-hourly scale, it also accounts for changes in storage in the canopy air space. As described in the companion paper (Longo et al., 2019), in ED-2.2, canopy air space storage is accounted for for energy, water, and carbon dioxide. While changes in storage are generally small in the seasonal or multi-annual scale (Fig. S8a,b), they are not negligible in the sub-daily scale. The relevance of the storage of $CO_2$ in the canopy air space has been long recognized by the eddy covariance community (e.g. Goulden et al., 1996; Bonal et al., 2008; Hayek et al., 2018b), but only rarely included in biosphere or land surface models. For example, the strong release of carbon dioxide in the early morning hours, resulting from the nighttime accumulation of respired $CO_2$, is well characterized by the model at both test sites (Fig. S8c,d). Accounting for this time lag between biologically-driven emission or uptake and the emissions to the free atmosphere is particularly important for benchmarking the model with the column-integrated measurements of the recently launched $CO_2$ Orbiting Carbon Observatory-3 OCO-3, which will provide samples at multiple times of the day (Eldering et al., 2017; Stavros et al., 2017).

## 4.3 Long-term, large-scale ecosystem dynamics

A key feature of the ED-2.2 model is the emergence of long-term, large-scale ecosystem composition, structure and function from spatially-localized, height-structured competition between individuals within the plant canopy. As seen in Fig. 10, the model's large-scale predictions are consistent with remote-sensing estimates of how AGB variability along key climatological gradients of incoming solar irradiance, rainfall, and dry-season severity. In addition, the regional pattern of AGB predicted by ED-2.2 reproduces several notable features present in the remotely-sensed based estimates of AGB (Fig. 7). In particular, the model captures the spatial extent of the Amazon forest, and reproduces two characteristic patterns of spatial variability in forest biomass, namely (i) the high biomass forests found in the Guiana Shield (near GYF) and in the area south of the TNF flux tower, and (ii) the area of lower biomass forest that runs east-to-west spanning the TNF and MDK sites.

The model's predictions of regional AGB also reveal important discrepancies. First, the model estimates are generally lower than the remote-sensing estimates of AGB in the drier savannas and xeric shrublands of central and Northeastern Brazil (Fig. 7-8). The low biomass estimates in drier regions is likely related to the simplified fire model used in the regional simulations . Following Moorcroft et al. (2001), fire occurrence within each climatological grid cell is controlled by a simple fixed soil moisture threshold, the area burned per year increases linearly as a function of the mean AGB within each grid cell, and no plants survive burn events. Although this simplified approach captures many patterns of fire regime in tropical South America (Fig. 9), it has important shortcomings in representing fire ignition mechanisms and the ecosystem's response to fire disturbance. For example, as previous work has shown (Cardoso et al., 2003; Cochrane, 2003; Andela et al., 2016), fire frequency, burn area, and fire severity are also strongly influenced by environmental factors in addition to soil moisture, such as proximity to roads and deforested areas. Moreover, the model does not account for size-related differences in fire survivorship and plant-functional diversity-related differences in fire survivorship arising from variation in plant traits such as bark-thickness-and re-sprouting ability (Brando et al., 2012; Trugman et al., 2018). Furthermore, the model simulations do not include plant functional types with adaptations for the semi-arid conditions typically observed in Northeastern Brazil. Such adaptations include smaller leaf size; internal water storage; modular, independent and redundant vascular systems; germination synchronized with rainfall; and Crassulacean Acid Metabolism (CAM) photosynthetic pathway (Cushman, 2001; Schenk et al., 2008; De Micco and Aronne, 2012). Incorporating these mechanisms that drive the ecosystem dynamics in drier areas could improve the model predictions outside tropical forests.

In the western Amazon, the model's predictions of AGB are generally higher than all three remote-sensing estimates, implying that the model is over-predicting AGB in this region (Fig. 7). ED-2.2 also tends to overestimate the high-biomass peak of the regional distribution of biomass (Fig. 8). One potential reason for the AGB over-estimation in the Western Amazon is the model's predicted dominance of late-successional plants over most of the Amazon region, whereas field observations indicate that forests in Western Amazon have higher stem turnover rates and lower wood densities than Eastern Amazon (Phillips et al., 2004; ter Steege et al., 2006). This has been linked to the fact that soils in Western Amazon have higher nutrient availability (Quesada et al., 2012), which was not accounted for in these ED-2.2 simulations. Moreover, the model assessment suggests that ED-2.2 overestimates growth rates (Fig. 13) and AGB (Fig. 8), despite evidence that ED-2.2 may be underestimating net

primary productivity (NPP; Fig. 5; Tab. 1; Fig. 11d). This result likely indicates that allocation to growth, maintenance, and reproduction may be biased. Trait and allometric data bases have considerably expanded over the past decade (e.g. Kattge et al., 2011; Falster et al., 2015; Keenan and Niinemets, 2016), with many of them have adopted open-access platforms. Incorporating these databases to develop and constrain allocation and maintenance costs of the functional groups in ED-2.2 could significantly reduce uncertainties in carbon stocks, productivity, and demographic rates. Finally, the regional simulation did not account for all types of anthropogenic disturbances from tropical forest degradation, which could also explain part of the overestimation by ED-2.2 compared to the remote-sensing estimates. Tropical forest degradation through selective logging, mining and low-intensity understory fires in the Amazon is pervasive along the arc of deforestation (Fig. 9b; Morton et al., 2013; Asner et al., 2013; Tyukavina et al., 2017) and is known to significantly deplete aboveground biomass (Berenguer et al., 2014; Longo et al., 2016; Rappaport et al., 2018).

The regional model simulation also qualitatively captures two disturbance-mediated relationships between canopy AGB and other tropical forest attributes observed in plot measurements (Fig. 11): the negative correlation between AGB and average stem mortality rates (Phillips et al., 2004; Johnson et al., 2016), and the positive correlation between AGB and average wood density found by Baker et al. (2004b). The fact that the model qualitatively captures the directional trend of both these two relationships is encouraging and suggests that the model's predictions of regional biomass trends are arising from mechanisms similar to those observed in nature. However, the magnitudes of the predicted relationships differ from the observations: for a given value of AGB, the model predicts a higher mean stem mortality rate and a higher mean wood density value than is observed in the plot measurements. The reasons for the differences in the magnitude of these relationships are, at present, unclear. However, in the case of the mean mortality-AGB relationship, the mismatch is likely related to the over-prediction of mortality rates seen in ED-2.2 (Fig. 11; 13). In addition, the predicted mortality rates at the two tropical forest sites showed nearly constant mortality rates during the period in which observed rates were available (Fig. 13). In ED-2.2, density-independent rates are modulated only by changes in functional composition (i.e., relative increase in abundance of pioneer plants), which did not occur in these simulations. Density-dependent mortality is associated with environmental conditions such as droughts and can drive the interannual variability of simulated mortality rates, yet these rates also nearly constant during this period. We interpret this result to be consistent, as the droughts of 2005 and 2010 were significant in the southwestern part of the Amazon (Lewis et al., 2011), but not at our study sites. The low sensitivity to 2005 and 2010 droughts is also supported by the observations, which did not show significant increases in mortality after 2005 and 2010 (Fig. 13); the increase in mortality at GYF between 2006 and 2008 is attributable to a substantial treefall event that affected one of the measurement plots in 2007. Nonetheless, the model does show sensitivity to historic droughts such as the 1992 and 1998 events in TNF (Longo et al., 2018) and the model response is consistent with anomalously high coarse woody debris found at the site when field inventory measurements began in 1999 (e.g. Pyle et al., 2008; Hayek et al., 2018a).

In this manuscript, we focused on assessing the model's ability to represent the dynamics of tropical forests, but previous studies have also shown that the model reasonably describes the dynamics of temperate ecosystems. For example, Medvigy and Moorcroft (2012) demonstrated that the model captures regional variation in both growth and mortality rates across forests of northeastern North America, especially when using the optimized parameters from Medvigy et al. (2009). Also, Miller et al.

(2016) applied ED-2.1 at Duke's Free Air Carbon dioxide Enrichment experiment site (conifer-dominated) and found that the model accurately predicts biomass changes over time for both ambient and elevated $CO_2$, and realistically characterizes the changes in gross and net primary productivity (NPP) as functions of stand age. This result is consistent with a previous model-intercomparison study, which found good agreement on observed and ED-2.1 modeled $CO_2$ fertilization effect at Duke, whereas the model predicted response to elevated $CO_2$ at Oak Ridge (broadleaf-dominated) was overestimated (De Kauwe et al., 2013). In contrast, in a millennium-long model inter-comparison study for Northeastern United States, the model overestimated both the magnitude of NPP and its variability as a function of rainfall and $CO_2$ when compared to tree-ring estimates (Rollinson et al., 2017), which indicates the need of constraining the model response for environmental conditions outside the current range. While these studies used versions of the model that contained many of the implementations of ED-2.2 described in the companion manuscript (Longo et al., 2019), future work using the ED-2.2 model in temperate ecosystems should critically evaluate processes that could not be assessed in our analysis, such as leaf phenology and processes related to snow dynamics. Such benchmarking efforts should be regarded as priorities in future studies.

## 5    Conclusions

Results from both observations and experimental studies have shown that plant diversity is an important determinant of terrestrial ecosystem function and how terrestrial ecosystems respond to environmental perturbation (Tilman, 1996; Gunderson, 2000; Cadotte et al., 2011; Mori et al., 2013; Hautier et al., 2015; Falster et al., 2017). Terrestrial ecosystem models have advanced significantly towards representing functional diversity over large regions over the past twenty years (Moorcroft et al., 2001; Medvigy and Moorcroft, 2012; Fisher et al., 2015, 2018), however their ability to represent complex, heterogeneous communities also depends on their ability to represent the heterogeneity of the environments where plants live and compete for resources. As we have shown in the companion paper (Longo et al., 2019), the model exhibits a high degree of conservation of energy, water, and carbon dioxide that is important in long-term ecosystem simulations, coupled biosphere-atmosphere modeling studies, and earth system model simulations. This is encouraging in light of a previous analysis that suggested a lack of energy conservation in the radiation schemes of several widely-used terrestrial biosphere models (Loew et al., 2014) and in the context of simulating the long-term dynamics of functionally diverse, vertically and horizontally heterogeneous plant canopy.

This detailed evaluation of the model performance in tropical South America, including an assessment of the separate components of the energy, water, and carbon cycles demonstrated the model's ability to represent multiple biophysical and biogeochemical mechanisms. The model dynamics are consistent with observations in most short-term fluxes of shortwave radiation (Fig. 1-2) and water (Fig. 4;S2), and even though there was significant overestimation of sensible heat fluxes, the model correctly characterized the average and the pattern of seasonality of canopy air space temperature (Fig. S2). The model represented realistic average magnitudes and light response curves of gross primary productivity (GPP) at GYF and TNF, albeit it did not capture the pattern of GPP seasonality at TNF (Fig. 5). Respiration showed the highest disagreement, both in terms of magnitude and seasonality, reflecting the uncertainties in representing respiration processes (Fig. 6; Table 1). In addition to the short-term comparisons with eddy covariance towers, ED-2.2 also showed good agreement with independent estimates

of aboveground biomass distribution at regional level in the tropics (Figs. 7-11) and the size distribution of the aggregated properties (Fig. 12), and reasonable magnitude of mortality rates and growth rates for TNF, an inland tropical forest site, although it significantly overestimated growth rates at GYF, a particularly nutrient-poor site (Fig. 13).

As pointed out in the companion model description manuscript (Longo et al., 2019), the ED-2.2 model continues to be developed. The ED-2.2 model evaluation presented here highlighted some short- and long-term processes that should be regarded as priorities for future developments. Better constrains in vegetation heat capacity could improve the quantification of energy storage and reduce biases in outgoing long wave radiation and sensible heat flux. Likewise, ecosystem respiration showed significant departures in magnitude and seasonality from site-level estimates. Formal optimization of parameters that 560 control respiration response to temperature and moisture, along with better description of the range of decomposition time scales may be required. Finally, the excessive tree growth rates identified in nutrient-poor sites may be addressed by the representation of biogeochemical cycles in tropical forests that accounts for nitrogen and phosphorus dynamics, which was recently implemented in ED-2.2 Medvigy et al. (2019) and could significant improve the characterization of the carbon cycle.

*Code and data availability.* The ED-2.2 software and further developments are publicly available. The most up-to-date source code, post- 565 processing R scripts, and an open discussion forum are available on https://github.com/EDmodel/ED2. The code described in this manuscript, along with a wiki-based technical manual, is stored as a release at https://github.com/mpaiao/ED2/releases/tag/rev-85 and permanently stored at https://dx.doi.org/10.5281/zenodo.2579481. Eddy covariance data from GYF and TNF sites are available at https://fluxnet.fluxdata.org. Forest inventory are available upon request to the data PI: GYF – D.B. (damien.bonal@inra.fr); TNF – P.B.d.C (pcamargo@cena.usp.br). The PGMF meteorological data set can be requested at http://hydrology.princeton.edu/data.pgf.php. The Oak Ridge National Laboratory 570 Distributed Active Archive Center provides access to the RADAMBRASIL (https://dx.doi.org/10.3334/ORNLDAAC/941), IGBP (http://dx.doi.org/10.3334/ORNLDAAC/565) and the photosynthetically active radiation (https://dx.doi.org/10.3334/ORNLDAAC/899) data sets. Remote-sensing biomass maps were obtained by contacting the corresponding authors (Saatchi et al., 2011; Baccini et al., 2012), and from http://lucid.wur.nl (Avitabile et al., 2016). CERES EBAF-Surface and TMPA-3B43 products are available at https://ceres.larc.nasa.gov and https://dx.doi.org/10.5067/TRMM/TMPA/MONTH/7, respectively. MODIS Leaf area index product (MCD15A2H, Collection 6) is available 575 at http://doi.org/10.5067/MODIS/MCD15A2H.006.

*Author contributions.* M.L., R.G.K., N.M.L., K.Z., R.L.B., S.C.W and P.R.M. designed the model assessment. M.L., R.G.K., N.M.L. and A.L.S.S. carried out the ED-2.2 simulations. D.B., B.B. P.B.C., M.N.H., S.C.S, R.d.S. and S.C.W contributed with observed meteorological, eddy flux, and forest inventory data. M.L., R.G.K., D.M.M., N.M.L., M.C.D., Y.K., A.L.S.S. and P.R.M wrote the paper.

*Competing interests.* The authors declare no competing interests.

*Acknowledgements.* The research was partially carried out at the Jet Propulsion Laboratory, California Institute of Technology, under a contract with the National Aeronautics and Space Administration. We thank the two anonymous reviewers, Miriam Johnston and Luciana Alves for suggestions that improved the manuscript; Luciana Alves, Bruce Daube, David Fitzjarrald, Elaine Gottlieb, Elizabeth Hammond-Pyle, Lucy Hutyra, Natalia Restrepo-Coupe, Raphael Tapajós, Scott Stark, and Kenia Wiedemann for the management and data processing; and Valerio Avitabile, Alessandro Baccini, and Sassan Saatchi for providing remote-sensing estimates of biomass. The model simulations

were carried out at the Odyssey cluster, supported by the FAS Division of Science, Research Computing Group at Harvard University. M.L. was supported by Conselho Nacional de Desenvolvimento Científico e Tecnológico (CNPq, grant 200686/2005-4), NASA Earth and Space Science Fellowship (NNX08AU95H) and National Science Foundation (NSF, grant OISE-0730305, Amazon-PIRE), São Paulo State Foundation (FAPESP, grant 2015/07227-6), and the NASA Postdoctoral Program, administered by Universities Space Research Association under contract with NASA. R.G.K was supported by a National Science Foundation Grant ATM-0449793 and National Aeronautics and

Space Administration Grant NNG06GD63G. A.L.S.S. was supported as a Giorgio Ruffolo Fellow in the Sustainability Science Program at Harvard University, for which support from Italy's Ministry for Environment, Land and Sea is gratefully acknowledged.

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
