# Peer review of "The biophysics, ecology, and biogeochemistry of functionally diverse, vertically- and horizontally-heterogeneous ecosystems: the Ecosystem Demography Model, version 2.2 — Part 2: Model evaluation for tropical South America"

_Geoscientific Model Development, 2019_

## Referee Comment (RC1) · Anonymous Referee #1 · 8 May 2019

The manuscript by Marcos Longo and colleagues performs a detailed evaluation of the ED-2.2 model (described in an accompanying discussion manuscript) for two sites in the Amazon and for the regional patterns simulated for the northern part of South America. The results are mostly presented in a comprehensive way and evaluate many different aspects of the model, both with regard to the short-term behaviour as "land surface scheme" and carbon cycle model and with regard to the long-term vegetation dynamics. It is hard to approach completeness in such an evaluation with all differ-

ent properties and processes simulated by a model of this complexity, but I think that the authors have presented a nice selection of results in which the most important processes as well as different types of processes are addressed.

These results, while not providing scientific novelties in themselves, provide a thorough evaluation of the model presented in the accompanying manuscript and provide good insight in the strengths and weaknesses of the current model implementation, and I think that these are presented in a balanced manner. I have some remarks about the way of describing and presenting some parts of the results that I would recommend the authors to address. With these adjustments, I expect the manuscript to be acceptable for publication in GMD.

I describe my remarks in more detail below, with some major issues and a list of smaller suggestions for edits and clarifications.

Major remarks:

- Evaluation of ED-2.2 is undertaken for the tropical forest in the Amazon in this study. Such a regional focus is understandable (and enough to be published separately), but as such the evaluation provided in the manuscript is for these tropical conditions specifically. The title of the manuscript could reflect the Amazon focus of this work. Also, it would be good to stress (p. 22, l. 21) that the model was evaluated for Amazon conditions specifically. E.g., this tests indeed "multiple biophysical and biogeochemical mechanisms", but also leaves many "mechanisms" that are typical for non-tropical conditions (e.g. those related to temperature-induced phenological changes or interactions with snow cover) unevaluated. I miss this aspect in the discussion and conclusion of the manuscript. The authors highlight that earlier versions of ED have been tested for other ecosystems as well (p. 21, l. 30ff), but do not discuss to what extent the current version of the model is expected to behave in a similar way or not.

- For the comparison of light extinction in the canopy (p. 3, l. 32; Fig. 2), it seems crucial that atmospheric conditions (e.g. the ratio of direct and diffuse light) are comparable to the average of the simulation period if you do not use the same days. Has this been tested? Even without detailed meteorological information for the time of measurements of the profile, I expect that there is some basic characterization of the weather conditions for those days that could be tested.

- It is unclear how the sensitivity of the model was tested (using average conditions from different forcing data sets, p. 4, l. 17), and how this relates to the regional simulations mentioned earlier. Are these separate simulations? Or are these meteorological drivers merely used to compute statistics to separate grid cells for determining relationships? In the case of the former, the description should be extended to describe the simulations properly, in the case of the latter, I am unsure why the authors have not used the existing model forcing to perform that separation?

- p. 5, l. 4: Yes, net radiation is partly determined by the incoming radiation (which is an input), but so is outgoing radiation of course. I expect the seasonality shown in Fig. 1 to be primarily determined by the seasonality of incoming radiation, and absolute deviations (in both outgoing and net radiation) are probably more informative for understanding model biases than the model's ability to represent the seasonal cycle.

- Fig. 2: Shading (confidence interval) appears to be missing in the figure.

- I like the summaries of functional relationships provided in Fig. 8 and 9. They are very informative to express the model's ability to represent spatial variations for the wider Amazon area.

- p. 22, l. 16: The ability of ED-2.2 to represent fine-scale heterogeneity is an interesting aspect, but was not evaluated in this study. Remarks to this would fit better in the accompanying "Part 1" paper than in this one.

Minor remarks:

- p. 1, l. 4: "excellent" could be removed here: You have verified the conservation of energy and other properties, not its excellency. In general, there is a tendency in

the manuscript to describe this conservation as an "excellent" property of the model – I trust that the authors are glad with that result, but conservation of properties is typically considered a technical prerequisite rather than a scientific breakthrough in model development.

- p. 3, l. 5: does "variability" here refer to spatial variability, temporal variability, or both?

- p. 3, l. 19: remove "the" in front of "each"

- p. 3, l. 19: Please clarify this sentence. I guess that GPP and Reco are "modelled statistically" from observed NEE, right? And have you tried to use NEE as well (next to the evaluations of GPP and Reco)? This is not clear to me at this point in the manuscript (it becomes more obvious when the results are presented).

- p. 4, l. 16: What was the spatial extent and spatial resolution of the regional simulation?

- p. 4, l. 29: The text is hard to follow here. Remove "the" in "the three remote sensing estimates," they are only introduced in the next sentence. And then, it seems that there are only two estimates introduced, whereas the third one (from the two sites only) cannot provide information on the spatial relationship. How was this used?

- p. 5, l. 4: The description of this sensitivity test is hard to follow – sensitivity to what is tested here, and what is the rationale of testing these different settings of the model?

- p. 5, l. 31: Apart from canopy air temperature, it would be interesting to learn whether surface temperature of the canopy is in agreement with observations (if those exist), to investigate whether the overestimated sensible heat fluxes are caused by too high temperatures.

- p. 12, l. 11: add "estimates of" between "remote sensing" and "biomass"

- p. 13, l. 5: Estimates of AGB from ED-2.2 appear to deviate substantially from observed ones, and the cloud of black points in Fig. 9a does not provide any confidence in

ED-2.2 to accurately predict AGB for different sites within the tropical forest. "Generally well characterized" seems a bit too optimistic for this.

- p. 20, l. 9: Sentence should probably read ". . . is accounted for for energy, water and carbon dioxide".

- p. 20, l. 10: remove "may"

- p. 21, l. 7: remove "significant"

- p. 22, l. 10: Correct "ecosystems"

- Fig. S3: remove "based on" from figure caption

[Figure]

---

## Referee Comment (RC2) · Anonymous Referee #2 · 11 Jun 2019

The authors present an interesting model evaluation of a new version of the Ecosystem Demography Model (version 2.2) for the Amazon region. They show strengths and weaknesses of the modelling approach and identify priorities for further model development. For the model evaluation, the authors use observational data from four specific sites, inventories and remote sensing and nicely present the comparison with the simulation results. Below, I have some comments and remarks that will hopefully help to improve the manuscript.

General comments:

- Title: The model evaluation has been done for the Amazon region, please state this in the title.

- The Methods section could profit from more structure and detail. First, although the manuscript has a companion paper that provides a detailed model description, I would recommend to give a summary of the model and a short overview over the relevant processes that are evaluated, in the beginning of the methods section. Additionally, it is to my opinion not fully clear, which datasets are used as forcing data and which are used for model evaluation (e.g. p.4, l.18ff).

- In the Methods section (p. 4, l. 17ff and p.5, l. 4) it is not clearly described how the model sensitivity was evaluated. Did you systematically vary different parameters or driver data?

- The assessment of forest function and structure is interesting and it is nice to see that the model can reproduce the functional relations (Fig. 9). I think it would be important to explain the reader that the mortality rate and wood density (e.g. Fig. 9b,c) are simulated by the model and not prescribed parameters, probably this could be done in the model summary in the methods section as suggested above. Regarding the unexplained bias in simulated mortality rates and wood density, it would also be interesting to see the relation between productivity (NPP) and mortality, which are probably the two dominant drivers for AGB. When looking at Fig. 5, it seems that (at plot level) GPP is underestimated and respiration is overestimated (Fig. 6), thus, NPP might be underestimated. With high mortality rates (Fig. 9) I wonder why biomass is overestimated (Fig. 7)?

Specific comments:

- Section 2.1: Which climate data were used to drive the model for the assessment of short-term fluxes? L. 18: What do you mean by "we aggregated the model results

to polygon level hourly averages"? Please explain the model setup more clearly (as in section 2.2). L. 19-22: The explanation in these sentences is difficult to understand. Did you compare NEE? Why is it relevant to compare "all times in which the net ecosystem productivity could be estimated..."?

- p. 4, l. 13: "We initialized soils with texture obtained from Quesada et al." What has exactly been done for the initialization?

- p. 4, l. 21ff: You assume a constant monthly evapotranspiration of 100 mm month-1 for calculating MCWD. This assumption is not valid in arid regions (L. 25). How reliable is the application of MCWD with the assumptions you made? Please also explain how the yearly MCWD is calculated, is it done for the hydrological year?

- p. 5, l. 4-5: "To evaluate the sensitivity of mortality and the growth rate": Sensitivity to what? Please specify.

- Page 5 line 13: How was outgoing shortwave radiation calculated by the model? Depending on the method, isn't outgoing-sw a proxy for LAI or FPC for which also site data should be available?

- p. 5, l. 19 and Fig. 2: Please define "TAI". The message of Fig. 2 is not clear. Is this related to LAI?

- p. 12, l. 5: Please move this sentence to the methods section and explain how the comparison of MODIS-LAI has been set up (which spatial resolution, for which time period, how did it match with information on deforestation considered in the simulations?)

- Figure 7: The symbols indicating the locations of the focus sites are very hard to see in the map.

- p. 13, l. 7: First time that fire is mentioned, leaving the reader a bit puzzled. It would be good to evaluate the occurrence of fire in more detail and to mention it in the methods section.

- Page 17, figure 17: ED2 shows (almost) constant mortality; density-independed and dependent. Why is it constant? Was there only background mortality occurring throughout the simulation period? What about the effects of the drought years (e.g. 2005 and 2010)? I would expect to see an effect in the model results.

---

## Author Comment (AC1) · 9 Jul 2019

**Author's response to reviews of "*The biophysics, ecology, and biogeochemistry of functionally diverse, vertically- and horizontally-heterogeneous ecosystems: the Ecosystem Demography Model, version 2.2 – Part 2: Model evaluation*"**

We would like to thank both reviewers for the comprehensive assessment and constructive comments on our first version of the manuscript.  Below we include the detailed, point-by-point response to each comment and question raised by each referee, and how we propose to address them in the revised manuscript. We have also included proposed text modifications following each of the referees' suggestions ("quoted blue text").

**Responses to Referee 1**

*R1 Comment 01: The manuscript by Marcos Longo and colleagues performs a detailed evaluation of the ED-2.2 model (described in an accompanying discussion manuscript) for two sites in the Amazon and for the regional patterns simulated for the northern part of South America. The results are mostly presented in a comprehensive way and evaluate many different aspects of the model, both with regard to the short-term behaviour as "land surface scheme" and carbon cycle model and with regard to the long-term vegetation dynamics. It is hard to approach completeness in such an evaluation with all different properties and processes simulated by a model of this complexity, but I think that the authors have presented a nice selection of results in which the most important processes as well as different types of processes are addressed.*

*These results, while not providing scientific novelties in themselves, provide a thorough evaluation of the model presented in the accompanying manuscript and provide good insight in the strengths and weaknesses of the current model implementation, and I think that these are presented in a balanced manner. I have some remarks about the way of describing and presenting some parts of the results that I would recommend the authors to address. With these adjustments, I expect the manuscript to be acceptable for publication in GMD.*

*I describe my remarks in more detail below, with some major issues and a list of smaller suggestions for edits and clarifications.*

> Response: We thank the Reviewer for their encouraging feedback.  Our goal in this manuscript was indeed to focus on evaluating multiple processes solved by ED-2.2 to provide a more comprehensive picture of which biophysical and ecological processes are well represented by the model, and which processes still need improvements in future developments.

*Major remarks:*

*R1 Comment 02: Evaluation of ED-2.2 is undertaken for the tropical forest in the Amazon in this study. Such a regional focus is understandable (and enough to be published separately), but as such the evaluation provided in the manuscript is for these tropical conditions specifically. The title of the manuscript could reflect the Amazon focus of this work. Also, it would be good to stress (p. 22, l. 21) that the model was evaluated for Amazon conditions specifically. E.g., this*

*tests indeed "multiple biophysical and biogeochemical mechanisms", but also leaves many "mechanisms" that are typical for non-tropical conditions (e.g. those related to temperature-induced phenological changes or interactions with snow cover) unevaluated. I miss this aspect in the discussion and conclusion of the manuscript. The authors highlight that earlier versions of ED have been tested for other ecosystems as well (p. 21, l. 30ff), but do not discuss to what extent the current version of the model is expected to behave in a similar way or not.*

Response: We agree with the Reviewer that we should make it clear in the title and abstract that the model evaluation is presented for tropical ecosystems. We will append "for tropical South America" in the title, and we will also modify the second text in the second paragraph of the *Conclusions*:

"This detailed evaluation of the model performance in tropical South America, including an assessment of the separate components of the energy, water, and carbon cycles demonstrated the model's ability to represent multiple biophysical and biogeochemical mechanisms."

We will also include a sentence in the discussion that highlights the need to further evaluate ED-2.2 in non-tropical biomes:

"While these studies used versions of the model that contained many of the implementations of ED- 2.2 described in the companion manuscript (Longo et al., 2019), future work using the ED-2.2 model in temperate ecosystems should critically evaluate processes that could not be assessed in our analysis, such as leaf phenology and processes related to snow dynamics. Such benchmarking efforts should be regarded as priorities in future studies."

*R1 Comment 03: For the comparison of light extinction in the canopy (p. 3, l. 32; Fig. 2), it seems crucial that atmospheric conditions (e.g. the ratio of direct and diffuse light) are comparable to the average of the simulation period if you do not use the same days. Has this been tested? Even without detailed meteorological information for the time of measurements of the profile, I expect that there is some basic characterization of the weather conditions for those days that could be tested.*

Response: Following the Reviewer's suggestion, we will include a new Supplemental Information figure that shows box-and-whisker plots of the daily averages of temperature, humidity, and net radiation from the nearest ABRACOS sites with such information, and the days we used to drive the ED-2.2 model. The range of temperature and humidity were similar between ABRACOS and LBA-MIP; for net radiation, daily averages are consistently higher during the LBA-MIP period, although the days we used from ED-2.2 predictions encompassed the entire range of values observed during the ABRACOS campaigns. For the diffuse-light only comparison (panels c and d), the characterization of weather conditions is not necessary: we are not comparing the absolute light extinction, but rather the normalized canopy radiation profile (i.e. divided by the incoming radiation measured immediately above the canopy), and we restrict the times from both the observations and simulations to be 100% diffuse light.

We will update the text in the Methods section to clarify the differences, the rationale for including the comparison under 100% diffuse radiation conditions, and point to the new SI figure:

"Although simulated days (LBA-MIP) include the entire range of daily averages of temperature, humidity, and net radiation observed during the ABRACOS campaign, the simulated period had significantly (t-test at 95% confidence) warmer average temperatures at MDK (0.67°C) and higher net radiation at both RJA (13.7 W m$^{-2}$) and MDK (18.5 W m$^{-2}$) (Fig. S1). Therefore, we also compared the predicted light extinction profile with observations for overcast hours, when the direct radiation (based on Weiss and Norman (1985) model) would be zero, when the model predictions of relative light extinction do not depend on the total incoming radiation (Longo et al., 2019)."

*R1 Comment 04*: *It is unclear how the sensitivity of the model was tested (using average conditions from different forcing data sets, p. 4, l. 17), and how this relates to the regional simulations mentioned earlier. Are these separate simulations? Or are these meteorological drivers merely used to compute statistics to separate grid cells for determining relationships? In the case of the former, the description should be extended to describe the simulations properly, in the case of the latter, I am unsure why the authors have not used the existing model forcing to perform that separation?*

*Response:* Neither CERES-EBAF nor TMPA-3B43 were used to drive simulations. These products are monthly averages, and ED-2.2 requires drivers with time resolution of the order of hours. We opted to use the remote-sensing data sets to make a completely independent assessment of the ED-2.2 simulations.

We propose to address the Reviewers comment (i) by clarifying that CERES-EBAF and TMPA-3B43 were used only to compute statistics, and (ii) by providing similar statistics based on the PGMF meteorological data that were used to drive the simulations as an additional figure in the Supplemental Information. We will change the text in the Methods Section:

"We examined the relationship between regional above-ground biomass (AGB) and light and water availability by comparing the model's AGB predictions against regional observations of these two quantities. Specifically for light availability, we compared the model predictions against two estimates of average annual shortwave radiation: one calculated from the PGMF shortwave measurements and that were used to drive the model simulations and second the annual average downwelling shortwave irradiance from the Clouds and the Earth's Radiant Energy System's Energy Balanced And Filled product (CERES-EBAF; Kato et al., 2013) between 2001 and 2017. Similarly, for rainfall, we compared the model's AGB prediction to annual rainfall estimates calculated from both the PGMF meteorological dataset and the average of annual precipitation calculated from the Tropical Rainfall Measurement Mission's (TRMM) Multi-Satellite Precipitation Analysis Product (TMPA-3B43 V7; TRMM, 2011; Liu et al., 2012) between 1998 and 2017."

: *p. 5, l. 4: Yes, net radiation is partly determined by the incoming radiation (which is an input), but so is outgoing radiation of course. I expect the seasonality shown in Fig. 1 to be primarily determined by the seasonality of incoming radiation, and absolute deviations (in both outgoing and net radiation) are probably more informative for understanding model biases than the model's ability to represent the seasonal cycle.*

> *Response:* We assumed that the Reviewer was referring to p. 5, l. 14.  We agree that the seasonality of outgoing radiation will also be modulated by the seasonality of incoming radiation, and we will modify the text to include this important caveat, by including the following sentence:
>
> "The seasonal variation of both $Q_{SW}\uparrow$ and $Q_{PAR}\uparrow$ is close to observations at both sites (Fig. 1a-d). This result is expected because the seasonality of outgoing radiation is strongly modulated by incoming radiation. Nevertheless, the differences between simulated and observed $Q_{SW}\uparrow$ and $Q_{PAR}\uparrow$ are also small: the bias in the annual…"

*R1 Comment 06*: *Fig. 2: Shading (confidence interval) appears to be missing in the figure.*

> Response: We will update the figure. In reality, the original figure had the confidence interval of the mean, which was nearly invisible because we are sampling from many days.  However, we  think it makes more sense to show the range of the daily averages instead, so the bands of the updated figure will show the 95% range around the mean for all days considered.  In the case of diffuse radiation, the model does not predict variability because all PFTs are assumed to have the same optical properties, and we are showing the normalized profile.  The updated Figure caption will contain these clarifications (also including Reviewer 2's suggestion – *R2 Comment 14*):
>
> "Figure 2. Comparison of downward photosynthetically active radiation profile relative to the top of the canopy. (a,b) Average of daily radiation as a function of cumulative tree area index (TAI); (c,d) average of times with diffuse radiation only for (a,c) Adolpho Ducke (MDK) and (b,d) Jaru (RJA). Shaded areas in the model correspond to the 95% range of daily averages. Simulated subsamples for diffuse radiation only do not show variability because all plant functional types are assumed to have the same optical properties.. Horizontal whiskers in observed values also correspond to the 95% confidence interval of daily and diffuse subsamples, whereas the vertical whiskers correspond to the 95% confidence interval of cumulative TAI at the points of measurement."

*R1 Comment 07*: *I like the summaries of functional relationships provided in Fig. 8 and 9. They are very informative to express the model's ability to represent spatial variations for the wider Amazon area.*

> Response: We are glad that the Reviewer finds Figures 8 and 9 interesting and informative about the model's regional-scale performance.

*R1 Comment 08*: *p. 22, l. 16: The ability of ED-2.2 to represent fine-scale heterogeneity is an interesting aspect, but was not evaluated in this study. Remarks to this would fit better in the accompanying "Part 1" paper than in this one.*

Response: As suggested, we will remove the following sentences regarding fine-scale heterogeneity from "Part 2". Similar sentences are present in Part 1.

~~"The ED-2.2 model accounts for this fine-scale heterogeneity by solving the energy, water, and carbon cycles for the different micro-environments in the plant community. The ED-2.2 model integrates biophysical, ecological and biogeochemical terrestrial ecosystem processes of heterogeneous landscapes on timescales ranging from minutes to centuries and on spatial scales ranging from individual plants to continental scales."~~

*Minor remarks:*

*R1 Comment 09: p. 1, l. 4: "excellent" could be removed here: You have verified the conservation of energy and other properties, not its excellency. In general, there is a tendency in the manuscript to describe this conservation as an "excellent" property of the model – I trust that the authors are glad with that result, but conservation of properties is typically considered a technical prerequisite rather than a scientific breakthrough in model development.*

Response:  We agree with the Reviewer that conservation of properties is a necessary condition and will revise the text to incorporate this statement. As suggested, we will remove the word "excellent" from both the *Introduction* and the *Conclusions* (p. 22, l. 19); However, we believe that the high degree of conservation exhibited by the model is an achievement that is worth noting given the findings of a previous analysis suggesting a lack of energy conservation in several terrestrial biosphere models (Loew et al. 2014) and given that the model is simulating the dynamics of diverse, spatially and horizontally heterogeneous, and temporally varying plant canopies over long timescales.

*R1 Comment 10: p. 3, l. 5: does "variability" here refer to spatial variability, temporal variability, or both?*

Response: We are referring to both, and we will clarify this in the text.

*R1 Comment 11: p. 3, l. 19: remove "the" in front of "each"*

Response: We will correct the text as suggested.

*R1 Comment 12: p. 3, l. 19: Please clarify this sentence. I guess that GPP and Reco are "modelled statistically" from observed NEE, right? And have you tried to use NEE as well (next to the evaluations of GPP and Reco)? This is not clear to me at this point in the manuscript (it becomes more obvious when the results are presented).*

Response: The Reviewer is correct, and we will modify the text accordingly.  Also, our original sentence incorrectly referred to NEE as net ecosystem productivity rather than net ecosystem exchange.  We will replace the original text with:

 "…when the each variable of interest was measured. Gross primary productivity (GPP) and ecosystem respiration ($R_{Eco}$) are not measured but statistically modeled from net ecosystem exchange (NEE), therefore we compared all times in which NEE could be estimated from tower observations."

*R1 Comment 13*: *p. 4, l. 16: What was the spatial extent and spatial resolution of the regional simulation?*

Response: The regional simulation was run over tropical South America (83°W–33°W; 18°S–13°N) at 1° spatial resolution.  We will include this information in the revised manuscript, and replace "Amazon ecoregion" with "tropical South America" as the domain includes areas outside the Amazon ecoregion.  The revised text (also incorporating suggestions from *R2 Comment 03*, *R2 Comment 10*, and *R2 Comment 17*) should read like:

"To evaluate the model ability to represent the long-term dynamics, we carried out multiple simulations intended to test the model's ability to describe regional variability as well as the structural and functional diversity of ecosystems in tropical South America.

First, to assess the model's ability to represent the distribution of biomass and leaf area index across tropical South America (83°W–33°W; 18°S–13°N), we carried out a 1−° resolution, 500-year-long regional simulation, starting from near-bare conditions (1400-1900) to produce the simulated potential vegetation. In these simulations, we assumed a constant treefall disturbance rate of 1.4%yr$^{-1}$ (Moorcroft et al., 2001), and allowed the occurrence of fire, using the original ED-1 implementation of fire disturbance model (Moorcroft et al., 2001). We then resumed the simulation in 1900, applying anthropogenic disturbance and ran the model until 2002, using a combination of land use transition matrices from Hurtt et al. (2006), nudged to match the initial conditions from Soares-Filho et al. (2006) in the Amazon. We specified  the soil texture for each grid cell using data from Quesada et al. (2011) for the Amazon, RADAMBRASIL (de Negreiros et al., 2009) for non-Amazonian areas of Brazil, and IGBP (Tempel et al., 1996) for non-Amazonian areas elsewhere. Soil texture characteristics (sand, silt, and clay content) are used to determine the pedotransfer function, hydraulic and thermal conductivities, and heat capacity of soils (Longo et al., 2019). For the meteorological forcing, we used the Princeton University Global Meteorological Forcing Dataset (PGMF, Sheffield et al., 2006) for 1969 to 2008, which was recycled multiple times to simulate a period equivalent to 1500 through 2002."

*R1 Comment 14*: *p. 4, l. 29: The text is hard to follow here. Remove "the" in "the three remote sensing estimates," they are only introduced in the next sentence. And then, it seems that there are only two estimates introduced, whereas the third one (from the two sites only) cannot provide information on the spatial relationship. How was this used?*

Response: We removed the "the" from this sentence. The confusion in this sentence was because the references to the remote sensing products were missing from the original sentence.  We were referring to Saatchi et al. (2011), Baccini et al. (2012), and Avitabile et al. (2016).  We will include these references in the revised manuscript.

*R1 Comment 15*: *p. 5, l. 4: The description of this sensitivity test is hard to follow – sensitivity to what is tested here, and what is the rationale of testing these different settings of the model?*

Response: We tested the sensitivity to factors that may affect the simulated dynamics yet are uncertain, but we agree that the text was not clear. We will include the following text in the revised manuscript:

"Several factors that are not well constrained for these sites may affect the simulations. For example, at both sites the meteorological drivers were not available for the period before the first inventory (in the case of TNF, meteorological data became available only two years after the first inventory). In addition, soil texture is known to be heterogeneous at local scale (e.g. Epron et al., 2006). Likewise, leaf phenology strategies within the local community can be diverse and heterogeneous (Bonal et al., 2000), and not known for the species included in the forest inventories. ED-2.2 model simulations are known to be modulated by such factors (e.g. Longo et al., 2018); to evaluate the sensitivity of mortality and growth rates to these factors, we carried out an ensemble of simulations…"

*R1 Comment 16: p. 5, l. 31: Apart from canopy air temperature, it would be interesting to learn whether surface temperature of the canopy is in agreement with observations (if those exist), to investigate whether the overestimated sensible heat fluxes are caused by too high temperatures.*

Response: We are not aware of long-term measurements of canopy temperature at the study sites. In the absence of canopy temperature measurements, we evaluated the outgoing longwave radiation ($Q_{LW}\uparrow$; Fig. S2), which should be closely related to canopy temperature in dense forests. As we pointed out in the original text, the model indeed overestimates $Q_{LW}\uparrow$ at both sites, especially during the afternoon, which suggests that canopy temperature may be overestimated. We will include the following sentence to the end of the paragraph:

"Therefore, the results suggest that the excessive sensible heat flux may be attributable to the model's overestimation of canopy temperature."

*R1 Comment 17: p. 12, l. 11: add "estimates of" between "remote sensing" and "biomass"*

Response: We will correct the text as suggested

*R1 Comment 18: p. 13, l. 5: Estimates of AGB from ED-2.2 appear to deviate substantially from observed ones, and the cloud of black points in Fig. 9a does not provide any confidence in ED-2.2 to accurately predict AGB for different sites within the tropical forest. "Generally well characterized" seems a bit too optimistic for this.*

Response: We will replace the original sentences with the following:

"The range and variability of biomass across the network are similar to the range and variability observed across most of the network (Fig. 10a). However, the model does not accurately predict the AGB of individual plots at the wettest sites (black dots in Fig. 10a). The model also predicts lower-than-observed biomass than at the drier sites located in Bolivia (red dots in Fig. 10a) because of frequent fires occurring in the model simulation."

*R1 Comment 19: p. 20, l. 9: Sentence should probably read ". . . is accounted for for energy, water and carbon dioxide".*

Response: We will rewrite the sentence accordingly.

*R1 Comment 20: p. 20, l. 10: remove "may"*

Response: As suggested, we will remove "may" from the sentence.

*R1 Comment 21: p. 21, l. 7: remove "significant"*

Response: As suggested, we will remove "significant" from the sentence.

*R1 Comment 22: p. 22, l. 10: Correct "ecosystems"*

Response: As suggested, we will correct this typographical error.

*R1 Comment 23: Fig. S3: remove "based on" from figure caption*

Response: As suggested, we will remove "based on" from the sentence.

**Responses to Referee 2**

*R2 Comment 01: The authors present an interesting model evaluation of a new version of the Ecosystem Demography Model (version 2.2) for the Amazon region. They show strengths and weaknesses of the modelling approach and identify priorities for further model development. For the model evaluation, the authors use observational data from four specific sites, inventories and remote sensing and nicely present the comparison with the simulation results. Below, I have some comments and remarks that will hopefully help to improve the manuscript.*

Response: We thank the Reviewer for the positive feedback. Our goal in this manuscript was indeed to show both the strengths and weaknesses of the current version and to guide which processes in the model should be prioritised in future model developments.

*General comments:*

*R2 Comment 02: Title: The model evaluation has been done for the Amazon region, please state this in the title.*

Response: We agree that the title should make it clear that it was a regional evaluation and accordingly we will include "for tropical South America" in the title of the revised manuscript.

*R2 Comment 03: The Methods section could profit from more structure and detail. First, although the manuscript has a companion paper that provides a detailed model description, I would recommend to give a summary of the model and a short overview over the relevant processes that are evaluated, in the beginning of the methods section. Additionally, it is to my opinion not fully clear, which datasets are used as forcing data and which are used for model evaluation (e.g. p.4, l.18ff).*

Response: In line with the Reviewer's suggestion, we plan to include the following paragraphs describing the model structure in the *Methods* section:

"Since its conception (Moorcroft et al., 2001; Hurtt et al., 2002), the plant community in the Ecosystem Demography model (ED) is represented by a hierarchical structure that accounts for both abiotic (e.g. soil texture and aspect) and biotic (e.g. age since last disturbance and disturbance types) effects on the vertical structure of canopy and the horizontal heterogeneity of such structures across the landscape. To characterize this sub-grid heterogeneity within the areas of interest and the changes of structure over time, ED solves a system of partial differential equations that describe growth, mortality and recruitment of individuals, as well as the changes in the plant community following disturbances. The current version of the model, ED-2.2, which is presented in a companion paper (Longo et al., 2019) builds on the developments by Medvigy et al. (2009) to quantify in detail the biophysical and biogeochemical cycles of energy, water, and carbon that fully accounts for the sub-grid heterogeneity of the above- and below-ground structure of the ecosystem.

In this manuscript, we evaluate a range of short-term (hours to seasons) biophysical and biogeochemical processes and long-term (years to centuries) demographic and ecological processes, with focus in tropical South America. Specifically, for the short-term processes, we used detailed information from eddy covariance tower sites in the Amazon to assess the model's ability to represent the canopy radiation transfer, evapotranspiration, sensible heat flux, temperature and water in sub-canopy pools (e.g. canopy air space and soils), and carbon fluxes (gross primary productivity and res-piration). For the long-term processes, we evaluated the model ability to represent the distribution of biomass and leaf area index across tropical South America, and relevant demographic characteristics such as the simulated forest struc- ture and demographic rates (e.g. mortality, growth). Importantly, we also tested the ecosystem response to environmental controls both for short- and long-term processes. Examples of such evaluations for short-term processes include the predicted response of gross primary productivity and respiration to environmental factors such as incoming radiation and soil moisture. For long-term processes, we tested how predicted carbon stocks were modulated by the regional variation of environmental controls such as rainfall and radiation, as well as emergent ecosystem properties such as the average mortality rate and average wood density, both of which are also predicted by the model."

Regarding the data sets used as forcing data and for model evaluation, the input data sets are described in the first paragraph of section 2.2, and the evaluation data sets are described in the second paragraph of this section. To remove ambiguity, we will change the wording in the text to clearly separate input drivers from evaluation data. See our response to *R1 Comment 04* for the proposed updates to the narrative.

*R2 Comment 04*: *In the Methods section (p. 4, l. 17ff and p.5, l. 4) it is not clearly described how the model sensitivity was evaluated. Did you systematically vary different parameters or driver data?*

Response: Regarding p. 4 l. 17, we assessed how the regional variation of biomass predicted by ED-2.2 was related to environmental drivers such as precipitation and radiation. We did not run multiple simulations. We will rewrite the sentence to eliminate this ambiguity. See our response to *R1 Comment 04* for the proposed updates to the narrative.

For p. 5, l. 4, we ran all the combinations of initial time, soil texture, and leaf phenology described in the text (48 simulations per site), but otherwise the simulations used the same parameters. We will update the text following our response to Reviewer 1 (*R1 Comment 15*), and also include the total number of simulations per site in the revised manuscript.

*R2 Comment 05: The assessment of forest function and structure is interesting and it is nice to see that the model can reproduce the functional relations (Fig. 9). I think it would be important to explain the reader that the mortality rate and wood density (e.g. Fig. 9b,c) are simulated by the model and not prescribed parameters, probably this could be done in the model summary in the methods section as suggested above.*

Response: The Reviewer is correct in pointing out that the community-level wood density and mortality are not prescribed parameters. We will rewrite the methods section to clarify this important point:

"To evaluate the model's ability to predict emergent properties, we used published values of biomass and mortality obtained from the RAINFOR field inventory network in the Amazon (Phillips et al., 2004; Baker et al., 2004a, b) and the results from long-term simulations near the field inventory sites (Levine et al., 2016). Specifically, we investigated the relationship between aboveground biomass and long-term average mortality rates, and community averaged wood density across these sites. In ED-2.2, typical wood density and background mortality rates are prescribed for each cohort, based on their plant functional type (PFT). However, the model allows for coexistence of PFTs that can be modulated by abiotic and biotic factors such as soil texture, climate, and direct competition for limiting resources (e.g. light and water) between PFTs (Moorcroft et al., 2001; Longo et al., 2019). Consequently, mortality rates and community-level wood density are not prescribed parameters; instead, these properties emerge from the population structure comprised by different plant functional types (PFTs)."

*R2 Comment 06: Regarding the unexplained bias in simulated mortality rates and wood density, it would also be interesting to see the relation between productivity (NPP) and mortality, which are probably the two dominant drivers for AGB. When looking at Fig. 5, it seems that (at plot level) GPP is underestimated and respiration is overestimated (Fig. 6), thus, NPP might be underestimated. With high mortality rates (Fig. 9) I wonder why biomass is overestimated (Fig. 7)?*

Response: The most likely explanation for the underestimated NPP and overestimated biomass is biases in the allocation of NPP to growth, reproduction, and maintenance. From Fig. 9, it is possible to see that ED-2.2 is over-estimating growth rates at the two

tested sites, which suggests that the model is underestimating maintenance costs of live tissues (e.g. leaf turnover, fine root turnover), underestimating allocation to reproduction (e.g. flowers, fruits and seeds), or both. As suggested, we will include a new panel in Fig. 9 (which will become Fig. 10 in the revised manuscript) that will compare the net primary productivity and mortality rates for a limited number of sites for which we had simulations and were able to find estimates of both quantities.  We will include the following text in the *Results* section:

"Fewer sites had estimates of both mortality rates and net primary productivity (NPP). However, the average predicted values at steady state (1.29 kgC $m^{-2}$ $yr^{-1}$ ) were slightly lower than values reported in the literature (1.35 kgC $m^{-2}$ $yr^{-1}$) for the selected sites (Fig. 10d). Consistent with previous observations in the Amazon (e.g. Quesada et al., 2012), both the ED-2.2 predictions indicate a positive correlation between NPP and mortality. However, this correlation is highly uncertain because of the limited number of plots (n = 7) and the large variability in both NPP and mortality across sites (Fig. 10d)."

Regarding the issues regarding underprediction of GPP, over-prediction of respiration and over-estimation of AGB, we agree with the Reviewer that reconciling these findings is important, and we will update the discussion to highlight the need to reduce uncertainties in carbon allocation:

"Moreover, the model assessment suggests that ED-2.2 overestimates growth rates (Fig. 12) and AGB (Fig. 9a), despite evidence that ED-2.2 may be underestimating net primary productivity (NPP; Fig. 5; Tab. 1; Fig. 10d). This result likely indicates that allocation to growth, maintenance, and reproduction may be biased. Trait and allometric data bases have considerably expanded over the past decade (e.g. Kattge et al., 2011; Falster et al., 2015; Keenan and Niinemets, 2016), with many of them have adopted open-access platforms. Incorporating these databases to develop and constrain allocation and maintenance costs of the functional groups in ED-2.2 could significantly reduce uncertainties in carbon stocks, productivity, and demographic rates."

*Specific comments:*

*Section 2.1:*

*R2 Comment 07: Which climate data were used to drive the model for the assessment of short-term fluxes?*

Response: We used the meteorological data from the eddy covariance towers. We will modify the text in the first paragraph of Section 2.1 to clarify this point:

"We used the meteorological variables measured at the eddy covariance tower sites to drive these ED-2.2 simulations. These variables included temperature, specific humidity, pressure, wind speed, incoming solar radiation, incoming longwave radiation and precipitation. All meteorological variables used as 20 inputs for ED-2.2 were gap filled, following Longo (2014)."

*R2 Comment 08: L. 18: What do you mean by "we aggregated the model results to polygon level hourly averages"? Please explain the model setup more clearly (as in section 2.2).*

Response: In line with the Reviewer's suggestion, we will revise the first sentence of this paragraph as follows to clarify its meaning:

"To ensure that model and observations at or near eddy covariance flux towers could be directly compared, and that the observed signal was strongly related to actual environment conditions, we saved model outputs every hour, and only used the model output for the hours when each variable of interest was measured. (…) Finally, in ED-2.2, we solve fluxes for each local plant community (patch) within the area of interest (polygon). Since eddy covariance towers provide flux measurements and estimates that are representative of the entire plant community, to assess ED-2.2 results against eddy covariance fluxes, we compared the tower estimates with the ED-2.2 polygon averages (i.e. area-weighted average of fluxes for all simulated patches)."

*R2 Comment 09: L. 19-22: The explanation in these sentences is difficult to understand. Did you compare NEE? Why is it relevant to compare "all times in which the net ecosystem productivity could be estimated. . ."?*

Response: Since our goal was to evaluate multiple processes in the model, we presented comparisons for gross primary productivity (GPP) and ecosystem respiration ($R_{Eco}$) separately . However, neither GPP nor $R_{Eco}$ are directly measured by the towers; instead, they are estimated from net ecosystem exchange (NEE), the difference between GPP and $R_{Eco}$). We intended to say that we only compared tower estimates of GPP and $R_{Eco}$ with ED-2.2 simulations for the hours in which NEE data were available. Please see our response to Reviewer 1 (*R1 Comment 12*) on how we propose to clarify this paragraph.

*R2 Comment 10: p. 4, l. 13: "We initialized soils with texture obtained from Quesada et al." What has exactly been done for the initialization?*

Response: We will rewrite these sentences to clarify the model setup and initialization procedures: our use of the word "initialization" was confusing in this context, since soil hydraulic and thermal characteristics are assumed to not change over time in ED-2.2. Please see the proposed changes in our response to Reviewer 1 (*R1 Comment 13*).

*R2 Comment 11: p. 4, l. 21ff: You assume a constant monthly evapotranspiration of 100 mm month-1 for calculating MCWD. This assumption is not valid in arid regions (L. 25). How reliable is the application of MCWD with the assumptions you made? Please also explain how the yearly MCWD is calculated, is it done for the hydrological year?*

Response: Our original Equation (1) was incorrect: it should have referred to the cumulative water deficit (CWD), not MCWD. CWD was integrated for each month continuously, i.e. previous water deficit is carried over to the next month, and can only be zero if rain exceeds CWD + ET. For MCWD, we selected the maximum CWD of each calendar year. This approach allows us to account for different precipitation seasonalities and interannual variability of dry-season length.

Regarding the application of our estimates to arid regions, we must clarify a few points. First, MCWD was used only for model assessment: ED-2.2 relies on soil moisture to parameterize ecological changes due to water stress (e.g. drought-induced mortality and fire frequency). Second, we chose to set $ET_0$ at 100 mm month$^{-1}$ because this value has

been frequently used in previous studies as a proxy for water stress in the Amazon region (e.g. Malhi et al. 2009; Lewis et al. 2011; Aragão et al. 2018). We agree with the Reviewer that the actual ET is likely to be lower in arid regions and consequently, actual MCWD is also likely lower than our estimates in these areas. However, our point was to verify that ED2 did not simulate forests in areas where forests should not exist. In this sense, our estimates of MCWD can be thought as MCWD if forests were present in a given location (forest-equivalent MCWD). If values of forest-equivalent MCWD become too high, then this is an indicator that forests should not exist in the place, which is what our analysis shows.

We will modify the methods section to clarify the calculation and explain the interpretation of MCWD for non-forested regions:

"For maximum cumulative water deficit (MCWD, mm), we assumed a constant monthly evapotranspiration ($ET_0$ = 100 mm mo$^{-1}$) and monthly precipitation (P , mm month$^{-1}$) from TMPA-3B43 or PGMF, following previous studies for the Amazon region (Malhi et al., 2009a; Lewis et al., 2011; Aragão et al., 2018). For any month $t$, we defined the cumulative water deficit (CWD, mm) to be:

$$CWD(t) = \min \{1200, \max [0, CWD(t - \Delta t) + ET_0 - P(t)]\}, \tag{1}$$

where $\Delta t$ = 1 month. This integration is continuously carried over the entire time series. For each calendar year, we defined MCWD as the maximum monthly value of CWD. Note that this estimate of MCWD assumes evapotranspiration values that are typical of tropical rainforests (i.e. $ET_0$ = 100mm) and therefore should be regarded as a forest-equivalent MCWD, not an actual measurement. High values of MCWD indicate regions where water losses would be too high to maintain forests. This baseline evapotranspiration is high for arid regions, and consequently we imposed a cap of 1200 mm on CWD to avoid extremely high deficits at the most arid regions, where precipitation is insufficient to bring the water deficit back to zero."

*R2 Comment 12: p. 5, l. 4-5: "To evaluate the sensitivity of mortality and the growth rate": Sensitivity to what? Please specify.*

Response: We tested the sensitivity to three factors known to be important, but that are uncertain for these two sites, namely, the absence of meteorological data prior to the first inventory, soil texture, and leaf phenology. See also our response to Reviewer 1 (*R1 Comment 15*) for the modifications we plan to implement in the revised manuscript.

*R2 Comment 13: Page 5 line 13: How was outgoing shortwave radiation calculated by the model? Depending on the method, isn't outgoing-sw a proxy for LAI or FPC for which also site data should be available?*

Response: Outgoing shortwave radiation ($Q_{SW}\uparrow$) is calculated using a multi-layer two-stream model for both photosynthetically active radiation (PAR) and near infrared (NIR) — this implementation is described in detail in part 1. Outgoing shortwave radiation is strongly related to leaf area index, especially in dense canopies such as the Amazon, but the relationship between $Q_{SW}\uparrow$ and leaf area index is modulated by the partition between direct and diffuse radiation, and soil moisture (albeit this effect is very small in

dense canopies).  Therefore, the translation between LAI and $Q_{SW}\uparrow$ is not direct, but mediated by environmental conditions.  We think that the comparison with tower measurements of $Q_{SW}\uparrow$ provides a more direct benchmarking and intend to keep the current comparison.

*R2 Comment 14: p. 5, l. 19 and Fig. 2: Please define "TAI". The message of Fig. 2 is not clear. Is this related to LAI?*

> Response: TAI stands for tree area index (also known as plant area index). TAI is related to LAI but it also accounts for branches and twigs.  We will include this explanation in the methods (end of section 2.1):

> "To evaluate the model ability to represent changes in light environment throughout the canopy, we compare the light profiles of model and observations as a function of the cumulative tree area index (TAI).  TAI is defined as the sum of leaf area index (LAI) and the branch wood area index (WAI).  TAI profiles were estimated from published data at RJA (Simon et al., 2005) and near MDK (McWilliam et al., 1993)."

> We will also include a reminder in the results section (3.1):

> "…the large uncertainties on the observed tree area index (TAI) profile and that the simulated years are not the same as the observations (Fig. 2a,b)…"

> We agree that the caption of Fig. 2 needs clarification. Please see our response to Reviewer 1 (*R1 Comment 06*) for the proposed changes.

*R2 Comment 15: p. 12, l. 5: Please move this sentence to the methods section and explain how the comparison of MODIS-LAI has been set up (which spatial resolution, for which time period, how did it match with information on deforestation considered in the simulations?)*

> Response: Following the Reviewer's suggestion, we will  the data set description from p.12, l.5 and the caption of Figure S4 to the methods and include the additional information that was missing.  We intend to add the following text to the methods:

> "We also compared the model's predictions of leaf area index (LAI) with estimates from the Moderate Resolution Imaging Spectroradiometer (MODIS, product MCD15A2H, Collection 6) (Yan et al., 2016). We used all cloud-free, high-quality data from MODIS-MCD15A2H that were available between August 2002 and July 2004.  We selected this period to reduce temporal — and land-use related — differences between the model simulation and MODIS-MCD15A2H, and we aggregated the average LAI to 1° resolution to be consistent with the ED-2.2 simulations."

> We will also modify the text in the *Results* section to the following:

> "We also found that ED-2.2 predicted a similar extent of LAI over the Amazon region when compared to MODIS estimates (Fig. S4)"

*R2 Comment 16: Figure 7: The symbols indicating the locations of the focus sites are very hard to see in the map.*

Response: As suggested, we will update Figures 7, S3, and S4, to improve visibility of the focus sites.

*R2 Comment 17: p. 13, l. 7: First time that fire is mentioned, leaving the reader a bit puzzled. It would be good to evaluate the occurrence of fire in more detail and to mention it in the methods section.*

Response: We will update the methods section to inform readers that the regional simulations included fire disturbance (see our suggestion for new paragraph in *R1 Comment 13*).

Also, following the Reviewer's suggestion, we will include an evaluation of the fire model with the Global Fire Emission Database (GFED4.1), which has annual burned area rates. We will include this information in the Methods section:

"Finally, to evaluate the model's predictions of fire regime across tropical South America, we compared the average fire disturbance rate over predicted by ED- 2.2 over one full cycle of the meteorological drivers (last 40 years of the simulation) with the Global Fire Emissions Database, version 4.1 (GFED4.1, Giglio et al., 2013; Randerson et al., 2018). Similarly to aboveground biomass and LAI, we aggregated GFED4.1 relative burned area to 1°; because GFED4.1 data has a short overlapping period with the PGMF drivers, we used all available years (1997–2015) to compute the average fire disturbance rate."

We are also going to include a new Figure in the main text (new Figure 8) that compares the predicted regional distribution of fire regime with GFED4.1, a new Supplemental Information Figure that shows the differences (new Figure S6), and include the following text in the Results section 3.2:

"The predicted transitions between the Amazon biome and the Cerrado (savanna biome in Central Brazil) and Los Llanos (grassland area in Venezuela) are driven by increased fire activity, consistent with the Global Fire Emission Database (GFED4.1 Giglio et al., 2013; Randerson et al., 2018) regional distribution of burned area ($\alpha$; Fig. 8). In addition, ED-2.2 correctly predicted higher burned area in the savanna region at the Brazil-Guyana-Venezuela border and low fire activity over most of the Amazon biome and in the Caatinga (low-biomass, semi-arid region in Northeastern Brazil) (Fig. 8). In contrast, ED-2.2 underestimated fire activity in El Beni (Bolivian savannahs, $\Delta\alpha = \alpha_{ED-2.2} - \alpha_{GFED4.1} = -6.8\%$ yr$^{-1}$) in the Colombian Los Llanos ($\Delta\alpha = -9.2\%$ yr$^{-1}$), while it overestimated fire disturbance in forest areas near the border of Brazil and Bolivia ($\Delta\alpha = 1.9\%$ yr$^{-1}$), the coastal areas of Brazilian states of Maranhão and Piauí, immediately east of the Amazon biome ($\Delta\alpha = 5.0\%$ yr$^{-1}$), and areas in Eastern Brazil south of 15°S ($\Delta\alpha = 3.7\%$ yr$^{-1}$) (Fig. 8; S6). Because the original fire model in ED-2.2 does not predict fire ignitions based on human activities (Moorcroft et al., 2001; Longo et al., 2019) ED-2.2 does not predict the burned areas along the arc of deforestation in the Southern and Eastern edges of the Amazon biome (Fig. 8a)."

Finally, we will update the text in the Discussion that highlights the need to improve the representation of fires in the model, to link with the results of the new Figure 8:

"Following Moorcroft et al. (2001), fire occurrence within each climatological grid cell is controlled by a simple fixed soil moisture threshold, the area burned per year increases linearly as a function of the mean AGB within each grid cell, and no plants survive burn events. Although this simplified approach captures many patterns of fire regime in tropical South America (Fig. 8), it has important shortcomings in representing fire ignition mechanisms and the ecosystem's response to fire disturbance. For example, as previous work has shown (e.g. Cardoso et al., 2003; Cochrane, 2003; Andela et al., 2016), fire frequency, burn area, and fire severity are also strongly influenced by environmental factors in addition to soil moisture, such as proximity to roads and deforested areas."

*R2 Comment 18: Page 17, figure 17: ED2 shows (almost) constant mortality; density-indepeded and dependent. Why is it constant? Was there only background mortality occurring throughout the simulation period? What about the effects of the drought years (e.g. 2005 and 2010)? I would expect to see an effect in the model results.*

Response: We assumed the Reviewer was referring to Figure 11. Indeed, during the period of forest inventory measurements (2004—2013 for GYF, and 1999—2011 for TNF), ED-2.2 simulations did not show any significant mortality, which was dominated by background mortality. We will include the following text in the discussion, for clarification:

[revised manuscript text omitted]

---

## Author Response (AR1)

Dear Dr. Müller,

Thank you and the two reviewers for the comprehensive assessment and constructive comments on our first version of the manuscript "*The biophysics, ecology, and biogeochemistry of functionally diverse, vertically- and horizontally-heterogeneous ecosystems: the Ecosystem Demography Model, version 2.2 --- Part 2: Model evaluation for tropical South America*" (gmd-2019-71, original title: "*The biophysics, ecology, and biogeochemistry of functionally diverse, vertically- and horizontally-heterogeneous ecosystems: the Ecosystem Demography Model, version 2.2 --- Part 2: Model evaluation*").

Below we include the detailed, point-by-point response to each comment and question raised by each referee. In our responses, line and page numbers correspond to the enclosed annotated manuscript and annotated supporting information. In the annotated files, blue text was added, and  was removed.

We used track changes along with GMD's LaTeX template, and we noticed that the line numbers on the left side of the text were incorrect in some pages. We tried to make the references consistent with the numbers shown in the margin whenever possible. We apologize for this inconvenience.

Sincerely,
Marcos Longo

**Responses to Referee 1**

*R1 Comment 01: The manuscript by Marcos Longo and colleagues performs a detailed evaluation of the ED-2.2 model (described in an accompanying discussion manuscript) for two sites in the Amazon and for the regional patterns simulated for the northern part of South America. The results are mostly presented in a comprehensive way and evaluate many different aspects of the model, both with regard to the short-term behaviour as "land surface scheme" and carbon cycle model and with regard to the long-term vegetation dynamics. It is hard to approach completeness in such an evaluation with all different properties and processes simulated by a model of this complexity, but I think that the authors have presented a nice selection of results in which the most important processes as well as different types of processes are addressed.*

*These results, while not providing scientific novelties in themselves, provide a thorough evaluation of the model presented in the accompanying manuscript and provide good insight in the strengths and weaknesses of the current model implementation, and I think that these are presented in a balanced manner. I have some remarks about the way of describing and presenting some parts of the results that I would recommend the authors to address. With these adjustments, I expect the manuscript to be acceptable for publication in GMD.*

*I describe my remarks in more detail below, with some major issues and a list of smaller suggestions for edits and clarifications.*

Response: We thank the Reviewer for their encouraging feedback. Our goal in this manuscript was indeed to focus on evaluating multiple processes solved by ED-2.2 to provide a more comprehensive picture of which biophysical and ecological processes are well represented by the model, and which processes still need improvements in future developments.

*Major remarks:*

*R1 Comment 02: Evaluation of ED-2.2 is undertaken for the tropical forest in the Amazon in this study. Such a regional focus is understandable (and enough to be published separately), but as such the evaluation provided in the manuscript is for these tropical conditions specifically. The title of the manuscript could reflect the Amazon focus of this work. Also, it would be good to stress (p. 22, l. 21) that the model was evaluated for Amazon conditions specifically. E.g., this tests indeed "multiple biophysical and biogeochemical mechanisms", but also leaves many "mechanisms" that are typical for non-tropical conditions (e.g. those related to temperature-induced phenological changes or interactions with snow cover) unevaluated. I miss this aspect in the discussion and conclusion of the manuscript. The authors highlight that earlier versions of ED have been tested for other ecosystems as well (p. 21, l. 30ff), but do not discuss to what extent the current version of the model is expected to behave in a similar way or not.*

Response: We agree with the Reviewer that we should make it clear in the title and abstract that the model evaluation is presented for tropical ecosystems. We appended "for tropical South America" in the title, and we modified the text in the *Conclusions* (p. 27, l. 27 of the annotated manuscript).

We also included a sentence in the discussion that highlights the need to further evaluate ED-2.2 in non-tropical biomes (p. 27, l. 5–9 of the annotated manuscript).

*R1 Comment 03: For the comparison of light extinction in the canopy (p. 3, l. 32; Fig. 2), it seems crucial that atmospheric conditions (e.g. the ratio of direct and diffuse light) are comparable to the average of the simulation period if you do not use the same days. Has this been tested? Even without detailed meteorological information for the time of measurements of the profile, I expect that there is some basic characterization of the weather conditions for those days that could be tested.*

Response: Following the Reviewer's suggestion, we included a new Supplemental Information figure (Figure S1) that shows box-and-whisker plots of the daily averages of temperature, humidity, and net radiation from the nearest ABRACOS sites with such information, and the days we used to drive the ED-2.2 model. The range of temperature and humidity were similar between ABRACOS and LBA-MIP; for net radiation, daily averages are consistently higher during the LBA-MIP period, although the days we used from ED-2.2 predictions encompassed the entire range of values observed during the ABRACOS campaigns. For the diffuse-light only comparison (panels c and d), the characterization of weather conditions is not necessary: we are not comparing the absolute light extinction, but rather the normalized canopy radiation profile (i.e. divided by the incoming radiation measured immediately above the canopy), and we restrict the times from both the observations and simulations to be 100% diffuse light.

We updated the text in the Methods section to clarify the differences, the rationale for including the comparison under 100% diffuse radiation conditions, and point to the new SI figure (p. 5, l. 2–8 of the annotated manuscript).

*R1 Comment 04: It is unclear how the sensitivity of the model was tested (using average conditions from different forcing data sets, p. 4, l. 17), and how this relates to the regional simulations mentioned earlier. Are these separate simulations? Or are these meteorological drivers merely used to compute statistics to separate grid cells for determining relationships? In the case of the former, the description should be extended to describe the simulations properly, in the case of the latter, I am unsure why the authors have not used the existing model forcing to perform that separation?*

> *Response:* Neither CERES-EBAF nor TMPA-3B43 were used to drive simulations. These products are monthly averages, and ED-2.2 requires drivers with time resolution of the order of hours. We opted to use the remote-sensing data sets to make a completely independent assessment of the ED-2.2 simulations.
>
> We addressed the Reviewer's comment (i) by clarifying that CERES-EBAF and TMPA-3B43 were used only to compute statistics, and (ii) by providing similar statistics based on the PGMF meteorological data that were used to drive the simulations as an additional figure in the Supplemental Information. We changed the text in the Methods Section (p. 6, l. 3–11 of the annotated manuscript).

*R1 Comment 05: p. 5, l. 4: Yes, net radiation is partly determined by the incoming radiation (which is an input), but so is outgoing radiation of course. I expect the seasonality shown in Fig. 1 to be primarily determined by the seasonality of incoming radiation, and absolute deviations (in both outgoing and net radiation) are probably more informative for understanding model biases than the model's ability to represent the seasonal cycle.*

> *Response:* We assumed that the Reviewer was referring to p. 5, l. 14. We agree that the seasonality of outgoing radiation will also be modulated by the seasonality of incoming radiation, and we modified the text to include this important caveat (p. 8, l. 3–5 of the annotated manuscript).

*R1 Comment 06: Fig. 2: Shading (confidence interval) appears to be missing in the figure.*

> Response: We will update the figure. In reality, the original figure had the confidence interval of the mean, which was nearly invisible because we are sampling from many days. However, we think it makes more sense to show the range of the daily averages instead, so the bands of the updated figure will show the 95% range around the mean for all days considered. In the case of diffuse radiation, the model does not predict variability because all PFTs are assumed to have the same optical properties, and we are showing the normalized profile. We updated the Figure 2 caption to include these clarifications, and also address Reviewer 2's suggestion (*R2 Comment 14*).

*R1 Comment 07: I like the summaries of functional relationships provided in Fig. 8 and 9. They are very informative to express the model's ability to represent spatial variations for the wider Amazon area.*

Response: We are glad that the Reviewer finds Figures 8 and 9 (Figures 9 and 10 in the annotated manuscript) interesting and informative about the model's regional-scale performance.

*R1 Comment 08: p. 22, l. 16: The ability of ED-2.2 to represent fine-scale heterogeneity is an interesting aspect, but was not evaluated in this study. Remarks to this would fit better in the accompanying "Part 1" paper than in this one.*

Response: As suggested, we removed the sentences regarding fine-scale heterogeneity from "Part 2" (p. 27, l. 17–20 of the annotated manuscript). Similar sentences are present in Part 1.

*Minor remarks:*

*R1 Comment 09: p. 1, l. 4: "excellent" could be removed here: You have verified the conservation of energy and other properties, not its excellency. In general, there is a tendency in the manuscript to describe this conservation as an "excellent" property of the model – I trust that the authors are glad with that result, but conservation of properties is typically considered a technical prerequisite rather than a scientific breakthrough in model development.*

Response:  We agree with the Reviewer that conservation of properties is a necessary condition and will revise the text to incorporate this statement. As suggested, we removed the word "excellent" from both the *Abstract* and the *Conclusions* (p.1, l.4 and p. 27, l. 21 of the annotated manuscript); However, we believe that the high degree of conservation exhibited by the model is an achievement that is worth noting given the findings of a previous analysis suggesting a lack of energy conservation in several terrestrial biosphere models (Loew et al. 2014) and given that the model is simulating the dynamics of diverse, spatially and horizontally heterogeneous, and temporally varying plant canopies over long timescales.

*R1 Comment 10: p. 3, l. 5: does "variability" here refer to spatial variability, temporal variability, or both?*

Response: We are referring to both, and we clarified this in the text (p. 3, l.6 of the annotated manuscript).

*R1 Comment 11: p. 3, l. 19: remove "the" in front of "each"*

Response: We corrected the text as suggested (p. 4, l. 11–12 of the annotated manuscript).

*R1 Comment 12: p. 3, l. 19: Please clarify this sentence. I guess that GPP and Reco are "modelled statistically" from observed NEE, right? And have you tried to use NEE as well (next to the evaluations of GPP and Reco)? This is not clear to me at this point in the manuscript (it becomes more obvious when the results are presented).*

Response: The Reviewer is correct, and we modified the text accordingly.  Also, our original sentence incorrectly referred to NEE as net ecosystem productivity rather than

net ecosystem exchange.  Please see the revised text (p. 4. l. 11–14 of the annotated manuscript).

*R1 Comment 13: p. 4, l. 16: What was the spatial extent and spatial resolution of the regional simulation?*

Response: The regional simulation was run over tropical South America (83°W–33°W; 18°S–13°N) at 1° spatial resolution.  We will include this information in the revised manuscript, and replace "Amazon ecoregion" with "tropical South America" as the domain includes areas outside the Amazon ecoregion.  We revised the text, which also incorporates suggestions from Reviewer 2 (*R2 Comment 03*, *R2 Comment 10*, and *R2 Comment 17*) (p. 5, l. 14–23 of the annotated manuscript).

*R1 Comment 14: p. 4, l. 29: The text is hard to follow here. Remove "the" in "the three remote sensing estimates," they are only introduced in the next sentence. And then, it seems that there are only two estimates introduced, whereas the third one (from the two sites only) cannot provide information on the spatial relationship. How was this used?*

Response: We removed the "the" from this sentence. The confusion in this sentence was because the references to the remote sensing products were missing from the original sentence.  We were referring to Saatchi et al. (2011), Baccini et al. (2012), and Avitabile et al. (2016).  We included these references in the revised manuscript (p. 6, l. 28–29 of the annotated manuscript).

*R1 Comment 15: p. 5, l. 4: The description of this sensitivity test is hard to follow – sensitivity to what is tested here, and what is the rationale of testing these different settings of the model?*

Response: We tested the sensitivity to factors that may affect the simulated dynamics yet are uncertain, but we agree that the text was not clear. We rewrote this part (p. 7, l. 19–26 of the annotated manuscript).

*R1 Comment 16: p. 5, l. 31: Apart from canopy air temperature, it would be interesting to learn whether surface temperature of the canopy is in agreement with observations (if those exist), to investigate whether the overestimated sensible heat fluxes are caused by too high temperatures.*

Response: We are not aware of long-term measurements of canopy temperature at the study sites.  In the absence of canopy temperature measurements, we evaluated the outgoing longwave radiation ($Q_{LW}\uparrow$; Fig. S2), which should be closely related to canopy temperature in dense forests.  As we pointed out in the original text (p. 10, l. 3–5 of the annotated manuscript), the model indeed overestimates $Q_{LW}\uparrow$ at both sites, especially during the afternoon, which suggests that canopy temperature may be overestimated (p. 10, l. 5–6 of the annotated manuscript).

*R1 Comment 17: p. 12, l. 11: add "estimates of" between "remote sensing" and "biomass"*

Response: We corrected the text as suggested (p. 16, l. 8 of the annotated manuscript).

*R1 Comment 18: p. 13, l. 5: Estimates of AGB from ED-2.2 appear to deviate substantially from observed ones, and the cloud of black points in Fig. 9a does not provide any confidence in*

*ED-2.2 to accurately predict AGB for different sites within the tropical forest. "Generally well characterized" seems a bit too optimistic for this.*

> Response: We revised the text to address the Reviewer's remarks (p. 17, l. 10–13 of the annotated manuscript).

*R1 Comment 19: p. 20, l. 9: Sentence should probably read ". . . is accounted for for energy, water and carbon dioxide".*

> Response: We rewrote the sentence accordingly (p. 24, l. 21 of the annotated manuscript).

*R1 Comment 20: p. 20, l. 10: remove "may"*

> Response: As suggested, we removed "may" from the sentence (p. 24, l. 22 of the annotated manuscript).

*R1 Comment 21: p. 21, l. 7: remove "significant"*

> Response: As suggested, we removed "significant" from the sentence (p. 25, l. 21 of the annotated manuscript).

*R1 Comment 22: p. 22, l. 10: Correct "ecosystems"*

> Response: As suggested, we corrected this typographical error (p. 27, l. 12 of the annotated manuscript).

*R1 Comment 23: Fig. S3: remove "based on" from figure caption*

> Response: As suggested, we removed "based on" from the sentence (Caption Fig. S4 of the annotated manuscript).

**Responses to Referee 2**

*R2 Comment 01: The authors present an interesting model evaluation of a new version of the Ecosystem Demography Model (version 2.2) for the Amazon region. They show strengths and weaknesses of the modelling approach and identify priorities for further model development. For the model evaluation, the authors use observational data from four specific sites, inventories and remote sensing and nicely present the comparison with the simulation results. Below, I have some comments and remarks that will hopefully help to improve the manuscript.*

> Response: We thank the Reviewer for the positive feedback. Our goal in this manuscript was indeed to show both the strengths and weaknesses of the current version and to guide which processes in the model should be prioritised in future model developments.

*General comments:*

*R2 Comment 02: Title: The model evaluation has been done for the Amazon region, please state this in the title.*

Response: We agree that the title should make it clear that it was a regional evaluation and accordingly we included "for tropical South America" in the title of the revised manuscript.

*R2 Comment 03: The Methods section could profit from more structure and detail. First, although the manuscript has a companion paper that provides a detailed model description, I would recommend to give a summary of the model and a short overview over the relevant processes that are evaluated, in the beginning of the methods section. Additionally, it is to my opinion not fully clear, which datasets are used as forcing data and which are used for model evaluation (e.g. p.4, l.18ff).*

Response: In line with the Reviewer's suggestion, we included two paragraphs describing the model structure in the *Methods* section (p. 3, l. 10–31 of the annotated manuscript).

Regarding the data sets used as forcing data and for model evaluation, the input data sets are described in the first paragraph of section 2.2, and the evaluation data sets are described in the second paragraph of this section. To remove ambiguity, we changed the wording in the text to clearly separate input drivers from evaluation data (p. 5, l. 14–23 of the annotated manuscript).

*R2 Comment 04: In the Methods section (p. 4, l. 17ff and p.5, l. 4) it is not clearly described how the model sensitivity was evaluated. Did you systematically vary different parameters or driver data?*

Response: Regarding p. 4 l. 17, we assessed how the regional variation of biomass predicted by ED-2.2 was related to environmental drivers such as precipitation and radiation. We did not run multiple simulations. We rewrote the sentence to eliminate this ambiguity (p. 6, l. 3–11 of the annotated manuscript).

For p. 5, l. 4, we ran all the combinations of initial time, soil texture, and leaf phenology described in the text (48 simulations per site), but otherwise the simulations used the same parameters. We updated the text following our response to Reviewer 1 (p. 7, l. 19–26 of the annotated manuscript), and also included the total number of simulations per site in the revised manuscript (p. 7, l. 26 of the annotated manuscript)..

*R2 Comment 05: The assessment of forest function and structure is interesting and it is nice to see that the model can reproduce the functional relations (Fig. 9). I think it would be important to explain the reader that the mortality rate and wood density (e.g. Fig. 9b,c) are simulated by the model and not prescribed parameters, probably this could be done in the model summary in the methods section as suggested above.*

Response: The Reviewer is correct in pointing out that the community-level wood density and mortality are not prescribed parameters. We rewrote the text in the methods section to clarify this important point (p. 7, l. 8–15 of the annotated manuscript).

*R2 Comment 06: Regarding the unexplained bias in simulated mortality rates and wood density, it would also be interesting to see the relation between productivity (NPP) and mortality, which are probably the two dominant drivers for AGB. When looking at Fig. 5, it seems that (at plot level) GPP is underestimated and respiration is overestimated (Fig. 6), thus, NPP might be underestimated. With high mortality rates (Fig. 9) I wonder why biomass is overestimated (Fig. 7)?*

Response: The most likely explanation for the underestimated NPP and overestimated biomass is biases in the allocation of NPP to growth, reproduction, and maintenance. From Fig. 9, it is possible to see that ED-2.2 is over-estimating growth rates at the two tested sites, which suggests that the model is underestimating maintenance costs of live tissues (e.g. leaf turnover, fine root turnover), underestimating allocation to reproduction (e.g. flowers, fruits and seeds), or both. As suggested, we included a new panel in Fig. 9 (Fig. 10d in the annotated manuscript) that compares the net primary productivity and mortality rates for a limited number of sites for which we had simulations and were able to find estimates of both quantities. We also included text analysing the results shown in Fig. 10d (p. 17, l. 19–24 of the annotated manuscript).

Regarding the issues regarding underprediction of GPP, over-prediction of respiration and over-estimation of AGB, we agree with the Reviewer that reconciling these findings is important, and we updated the discussion to highlight the need to reduce uncertainties in carbon allocation (p. 25, l. 28–34 of the annotated manuscript).

*Specific comments:*

*Section 2.1:*

*R2 Comment 07: Which climate data were used to drive the model for the assessment of short-term fluxes?*

Response: We used the meteorological data from the eddy covariance towers. We modified the text in the first paragraph of Section 2.1 to clarify this point (p. 4, l. 4–7 of the annotated manuscript).

*R2 Comment 08: L. 18: What do you mean by "we aggregated the model results to polygon level hourly averages"? Please explain the model setup more clearly (as in section 2.2).*

Response: In line with the Reviewer's suggestion, we revised the first sentence of this paragraph as follows to clarify its meaning (p. 4, l. 9–19 of the annotated manuscript)..

*R2 Comment 09: L. 19-22: The explanation in these sentences is difficult to understand. Did you compare NEE? Why is it relevant to compare "all times in which the net ecosystem productivity could be estimated. . ."?*

Response: Since our goal was to evaluate multiple processes in the model, we presented comparisons for gross primary productivity (GPP) and ecosystem respiration ($R_{Eco}$) separately . However, neither GPP nor $R_{Eco}$ are directly measured by the towers; instead, they are estimated from net ecosystem exchange (NEE), the difference between GPP and $R_{Eco}$). We intended to say that we only compared tower estimates of GPP and $R_{Eco}$ with

ED-2.2 simulations for the hours in which NEE data were available.  Please see the revised text (p. 4. l. 11–14 of the annotated manuscript).

*R2 Comment 10*: *p. 4, l. 13: "We initialized soils with texture obtained from Quesada et al." What has exactly been done for the initialization?*

Response: We rewrote these sentences to clarify the model setup and initialization procedures (p. 5, l. 14–23 of the annotated manuscript): our use of the word "initialization" was confusing in this context, since soil hydraulic and thermal characteristics are assumed to not change over time in ED-2.2.

*R2 Comment 11*: *p. 4, l. 21ff: You assume a constant monthly evapotranspiration of 100 mm month-1 for calculating MCWD. This assumption is not valid in arid regions (L. 25). How reliable is the application of MCWD with the assumptions you made? Please also explain how the yearly MCWD is calculated, is it done for the hydrological year?*

Response: Our original Equation (1) was incorrect: it should have referred to the cumulative water deficit (CWD), not MCWD.  CWD was integrated for each month continuously, i.e. previous water deficit is carried over to the next month, and can only be zero if rain exceeds CWD + ET.   For MCWD, we selected the maximum CWD of each calendar year.  This approach allows us to account for different precipitation seasonalities and interannual variability of dry-season length.

Regarding the application of our estimates to arid regions, we must clarify a few points. First, MCWD was used only for model assessment: ED-2.2 relies on soil moisture to parameterize ecological changes due to water stress (e.g. drought-induced mortality and fire frequency). Second,  we chose to set $ET_0$ at 100 mm month$^{-1}$ because this value has been frequently used in previous studies as a proxy for water stress in the Amazon region (e.g. Malhi et al. 2009; Lewis et al. 2011; Aragão et al. 2018).  We agree with the Reviewer that the actual ET is likely to be lower in arid regions and consequently, actual MCWD is also likely lower than our estimates in these areas.  However, our point was to verify that ED2 did not simulate forests in areas where forests should not exist.  In this sense, our estimates of MCWD can be thought as MCWD if forests were present in a given location (forest-equivalent MCWD).  If values of forest-equivalent MCWD become too high, then this is an indicator that forests should not exist in the place, which is what our analysis shows.

We modified the methods section to clarify the calculation and explain the interpretation of MCWD for non-forested regions (p. 6. l. 11–24 of the annotated manuscript).

*R2 Comment 12*: *p. 5, l. 4-5: "To evaluate the sensitivity of mortality and the growth rate": Sensitivity to what? Please specify.*

Response:  We tested the sensitivity to three factors known to be important, but that are uncertain for these two sites, namely, the absence of meteorological data prior to the first inventory, soil texture, and leaf phenology. We rewrote the text to clarify these points (p. 7, l. 19–26 of the annotated manuscript).

*R2 Comment 13*: *Page 5 line 13: How was outgoing shortwave radiation calculated by the model? Depending on the method, isn't outgoing-sw a proxy for LAI or FPC for which also site data should be available?*

Response: Outgoing shortwave radiation ($Q_{SW}\uparrow$) is calculated using a multi-layer two-stream model for both photosynthetically active radiation (PAR) and near infrared (NIR) — this implementation is described in detail in part 1. Outgoing shortwave radiation is strongly related to leaf area index, especially in dense canopies such as the Amazon, but the relationship between $Q_{SW}\uparrow$ and leaf area index is modulated by the partition between direct and diffuse radiation, and soil moisture (albeit this effect is very small in dense canopies). Therefore, the translation between LAI and $Q_{SW}\uparrow$ is not direct, but mediated by environmental conditions. We think that the comparison with tower measurements of $Q_{SW}\uparrow$ provides a more direct benchmarking and kept the original comparison.

*R2 Comment 14*: *p. 5, l. 19 and Fig. 2: Please define "TAI". The message of Fig. 2 is not clear. Is this related to LAI?*

Response: TAI stands for tree area index (also known as plant area index). TAI is related to LAI but it also accounts for branches and twigs. We included this explanation in the Methods section (p. 5, l. 10–11 of the annotated manuscript), and a reminder in the Results section (p. 9, l. "–1" of the annotated manuscript). We also agree that the caption of Fig. 2 needed clarification and rewrote the text.

*R2 Comment 15*: *p. 12, l. 5: Please move this sentence to the methods section and explain how the comparison of MODIS-LAI has been set up (which spatial resolution, for which time period, how did it match with information on deforestation considered in the simulations?)*

Response: Following the Reviewer's suggestion, we moved the data set description to the methods (p. 6, l.29–33 of the annotated manuscript) and included the additional information that was missing. We also modified the text in the *Results* section (p. 14, l.8–9 of the annotated manuscript).

*R2 Comment 16*: *Figure 7: The symbols indicating the locations of the focus sites are very hard to see in the map.*

Response: As suggested, we updated the figures to improve visibility of the focus sites (Fig. 7, 8, S4, S5, and S6 of the annotated manuscript).

*R2 Comment 17*: *p. 13, l. 7: First time that fire is mentioned, leaving the reader a bit puzzled. It would be good to evaluate the occurrence of fire in more detail and to mention it in the methods section.*

Response: We updated the methods section to inform readers that the regional simulations included fire disturbance (p. 5, l.29–33 of the annotated manuscript).

Also, following the Reviewer's suggestion, we included an evaluation of the fire model with the Global Fire Emission Database (GFED4.1), which has annual burned area rates. We added this information in the Methods section (p. 7, l.0–5 of the annotated

manuscript).  We also included a new Figure 8 in the main text that compares the predicted regional distribution of fire regime with GFED4.1, a new Supplemental Information Figure that shows the differences (new Figure S6). Moreover,  we added text describing the comparison in the Results section (p. 14. l.11–p. 16 l.1 of the annotated manuscript).  Finally, we updated the text in the Discussion that highlights the need to improve the representation of fires in the model, to link with the results of the new Figure 8 (p. 25, l.10–12 of the annotated manuscript).

*R2 Comment 18: Page 17, figure 17: ED2 shows (almost) constant mortality; density-independed and dependent. Why is it constant? Was there only background mortality occurring throughout the simulation period? What about the effects of the drought years (e.g. 2005 and 2010)? I would expect to see an effect in the model results.*

Response: We assumed the Reviewer was referring to Figure 11.  Indeed, during the period of forest inventory measurements (2004—2013 for GYF, and 1999—2011 for TNF), ED-2.2 simulations did not show any significant mortality, which was dominated by background mortality.  We included additional text in the discussion, for clarification (p. 26, l.14–27 of the annotated manuscript).

[revised manuscript text omitted]

Figure S1: Box-and-whisker plots of the daily averages of (a) temperature, (b) specific humidity, and (c) net radiation of sites Jaru Biological Reserve (RJA) and the nearest sites from Adolpho Ducke Forest Reserve (MDK). The periods of comparison correspond to the Anglo-Brazilian Climate Observation Study (ABRACOS Cabral et al., 1996; Tomasella et al., 2008), and the data from the Large-Scale Biosphere-Atmosphere Experiment in Amazonia Data Model Intercomparison Project (de Gonçalves et al., 2013; de Gonçalves et al., 2013, LBA-MIP). For RJA, we used ABRACOS data from Aug-Sep 1992 and Apr-Jun 1993; for MDK, we used data from Fazenda Dimona (Jul-Aug 1991). For the LBA-MIP, we used the same days of year with ABRACOS measurements, from sites RJA (1999-2002) and K34 (1999-2006). Boxes correspond to the interquartile range, whiskers encompass data within 1.5 times the interquartile range, and the black line in the boxes are the median.

[Figure]

Figure S2: Fortnightly means of the (a,b) mean canopy air space temperature and (c,d) mean canopy air space specific humidity, obtained from the model and from the vertical average of the (a,c) GYF and (b,d) TNF tower measurements. Data for the TNF tower are available between 2002 and 2005 (Hutyra et al., 2008). Bands are the 95% confidence interval of means obtaining from bootstrapping, and rectangles in the background correspond to the site's climatological dry season. The missing periods are due to insufficient data of all times of the day to generate the fortnightly averages.

[Figure]

Figure S3: (a-b) Kernel density estimate of daily averages of outgoing thermal infrared (longwave) irradiance predicted by the model and measured at the towers for sites (a) GYF and (b) TNF. Only days with no gaps in observations were used to estimate kernel density for both observations and model. (c-d) Box-and-whisker plot of outgoing thermal infrared irradiance from ED-2.2 and observations, organized by time of day, for sites (c) GYF and (d) TNF. The model distribution includes only the times for which a corresponding observation existed.

[Figure]

Figure S4: (a) Regional aboveground biomass obtained from ED-2.2, and difference between ED-2.2 and  remote-sensing estimates from (b) Saatchi et al. (2011), (c) Baccini et al. (2012), and (d) Avitabile et al. (2016). Remote-sensing maps were aggregated to 1° resolution. Positive (negative) values in (b-d) mean that ED-2.2 predicted higher (lower) aboveground biomass than remote sensing estimates. The location of focus sites of Paracou (GYF, □) and Tapajós (TNF, ◇), and the sites used for radiation profile evaluation: Ducke (MDK, ⋇△), and Jaru (RJA, +▽) are shown for reference. Thick contour is the domain of the Amazon biome, and thin contours are the political borders. **(Figure was updated, following R2 suggestion.)**

[Figure]

Figure S5: Comparison of leaf area index from (a) ED-2.2, and (b) based on remote-sensing estimates from Moderate Resolution Imaging Spectroradiometer (MODIS), product MCD15A2H, Collection 6 (Yan et al., 2016); (c) Difference between ED-2.2 and the MODIS-MCD15A2H product. Estimates from MODIS-MCD15A2H used all cloud-free, high-quality data available between August 2002 and July 2004. The location of focus sites of Paracou (GYF, □) and Tapajós (TNF, ◊), and the sites used for radiation profile evaluation: Ducke (MDK, ⨉△), and Jaru (RJA, ╀▽) are shown for reference. Thick contour is the domain of the Amazon biome, and thin contours are the political borders. **(Figure was updated, following R2 suggestion.)**

[Figure]

Figure S6: (a) Regional burned area obtained from ED-2.2, and (b) difference between ED-2.2 and the Global Fire Emission database (GFED4.1 Giglio et al., 2013; Randerson et al., 2018) burned area product (1997–2015 average). Remote-sensing maps were aggregated to 1° resolution. Positive (negative) values in (b-d) mean that ED-2.2 predicted higher (lower) aboveground biomass than GFED4.,1 estimates. The location of focus sites of Paracou (GYF, □) and Tapajós (TNF, ◇), and the sites used for radiation profile evaluation: Ducke (MDK, △), and Jaru (RJA, ▽) are shown for reference. Thick contour is the domain of the Amazon biome, and thin contours are the political borders.

[Figure]

Figure S7: Average biomass predicted by ED-2.2 and based on remote-sensing maps, aggregated by annual averages of (a) downwelling shortwave irradiance; (b) mean annual precipitation; (c) maximum cumulative water deficit. For each annual average of environmental properties, grid points were grouped into 20 quantile bins: points represent the average within each bin, and shaded area corresponds to the 90% quantile range within each bin. Data source for the annual means: downwelling shortwave irradiance and precipitation Princeton Global Meteorological Forcing (PGMF; Sheffield et al., 2006) (1969-2008), which were also used as drivers for the ED-2.2 simulation; maximum cumulative water deficit was based on the same approach as Malhi et al. (2009a), using the PGMF precipitation.

[Figure]

Figure S8: Mean annual cycle from fortnightly means of change in $CO_2$ storage in the canopy air space for (a) GYF and (b) TNF. Bands are the 95% confidence interval of means, and rectangles in the background correspond to the site's climatological dry season. Box plot of change in $CO_2$ storage in the canopy air space aggregated by time of day for all hours with available data for (c) GYF and (d) TNF.

[Figure]

Figure S9: Kernel density estimate of daily averages of (a-b) volumetric soil moisture and (c-d) relative soil moisture at (a,c) GYF (depth 53 cm) and (b,d) TNF (depth 20 cm). Data for GYF were collected at the tower site (2007-2012), and data for TNF were collected at a site located 15 km south of the tower between 2001 and 2003 and available from Miller et al. (2009) and de Gonçalves et al. (2013). Vertical dashed lines in panels (a) and (b) correspond to the residual, permanent wilting point, field capacity and porosity calculated by ED-2.2 using the reference soil texture at each site: (56% sand; 9% silt; 35% clay) for GYF (Bonal et al., 2008), and (39% sand; 2% silt; 59% clay) for TNF .(Nepstad et al., 2002) Soil wetness was defined as the relative water content between residual (0%) and porosity (100%).

**S1 Derivation of the respiration estimates at Guyaflux (GYF)**

To obtain respiration estimates at Guyaflux (GYF), we follow the same methodology described by Malhi et al. (2009b), and used the same approach to obtain the standard error estimates. Whenever possible, we used published estimates of the different terms of the respiration terms.

- *Leaf respiration*. Individual-level leaf respiration was measured and reported by Stahl (2010). Leaf-level measurements were scaled to ecosystem scale using LAI-2000 measurements also by Stahl (2010). Following Malhi et al. (2009b) and Lloyd et al. (2010), the value was scaled down by 34% to account for diurnal down-regulation of dark respiration.

- *Stem respiration*. Individual-level stem respiration was measured and reported by Stahl (2010); Stahl et al. (2011). Values from both *terra firme* and seasonably flooded forest were included. Wet season and dry season values were weighted by the season length in 2008, also reported by Stahl et al. (2011). Although Stahl et al. (2011) had scaled to stand level, we recalculated their estimate to be consistent with Malhi et al. (2009b), by finding the mean stem area index from the forest inventory, following Chambers et al. (2000, 2004). The stem area index was $1.07 \pm 0.03$ m$^2$ m$^{-2}$.

- *Total soil respiration*. Total soil respiration was reported by Epron et al. (2006) and Bréchet et al. (2011). Following Malhi et al. (2009b), their estimates were averaged using an weighting factor equivalent to the sampling area times the square root of the duration of the measurments.

- *Coarse woody debris respiration*. Coarse woody debris (CWD) respiration was reported by Rowland et al. (2013). Their estimates of coarse woody debris respiration were scaled by soil water content, and thus these estimates are not entirely observed.

- *Soil Heterotrophic respiration*. This term was estimated by Bréchet (2009), using trenching method.

- *Root respiration*. This term was assumed to be the difference between total soil respiration and soil heterotrophic respiration, following Malhi et al. (2009b).

- *Aggregated respiration terms*. Heterotrophic respiration (soil + CWD), autotrophic respiration (leaf + root + stem), and ecosystem respiration (heterotrophic + autotrophic) were defined

as the sum of the terms, as defined above. Uncertainties were assumed independent, and propagated similarly to Malhi et al. (2009b).